# How does turbulence affect wake development in floating wind turbines? Some insights from comparative LES simulations and wind tunnel experiments

Leonardo Pagamonci[1], Francesco Papi[1], Gabriel Cojocaru[2], Marco Belloli[3], Alessandro Bianchini[1]

[1] Department of Industrial Engineering (DIEF), Università degli Studi di Firenze, Firenze, 50139, Italy
[2] Convergent Science GmbH, Hauptstrasse 10, 4040 Linz, Austria
[3] Department of Mechanics, Politecnico di Milano, Milano, 20156, Italy

*Correspondence to*: Francesco Papi (fr.papi@unifi.it)

**Abstract.** Research is flourishing on how to model, mitigate, or even try to exploit the complex motions floating offshore wind turbines (FOWTs) are subjected to due to the combined loading from wind, waves, currents, and buoyancy effects. While preliminary studies made use of simplified inflows to focus attention on blade-flow interaction, recent evidence suggests that the impact of realistic inflows can be much larger than expected. The present study presents a critical analysis aimed at quantifying to what extent turbulence characteristics affect the wake structures of a floating turbine undergoing large motions. Numerical CFD simulations, using a Large Eddy Simulation (LES) approach coupled with an Actuator Line Method for the rotor, are benchmarked against wind tunnel experimental data from the first campaign of the NETTUNO project on a scaled rotor that was tested both in static conditions and when oscillating in pitch. A comparative analysis of the results at different turbulence levels first confirmed that, whenever idealized flows with no significant turbulence are considered, platform motion in FOWTs indeed leads to the creation of induced flow structures in the wake that dominate its development and the vortex breakdown in comparison to bottom-fixed cases. More interestingly, analyses show on the other hand that, whenever realistic turbulence comes to play, only small gains in terms of wake recovery are noticed in FOWTs in comparison to bottom-fixed turbines, suggesting the absence of superposition effects between inflow and platform motion, with inflow turbulence contributing significantly to dissipating the structures induced by turbine oscillation. Finally, as an ancillary outcome of the study, evidence provided by LES high-fidelity simulations were used to understand to what extent a less computationally-intensive URANS approach can be used to study the impact of realistic turbulence. In particular, an innovative URANS approach featuring improved inflow boundary conditions proved to yield consistent results if mean wake profiles are considered.

## 1 Introduction

Floating offshore wind turbines (FOWTs) are seen as the enabling technology to boost wind energy production worldwide (Musial et al., 2020) as they will allow exploiting windy sites offshore even if characterized by significant water depths. While

industry has started developing the first large-scale floating wind farms, technical solutions for such projects are inevitably conservative as still mostly adapted from bottom-fixed case studies. On the other hand, the research community is working in parallel to progress in technology and develop more advanced solutions for a massive but sustainable deployment in years to come (Veers et al., 2022). Challenges posed by FOWTs are indeed many at all system levels (Veers et al., 2022). Due to their deployment in open sea and the unprecedented dimensions of modern rotors, these turbines will have to face metocean

conditions that can be extremely variable and include extreme events (McCann, 2016); moreover, blades can now exceed the atmospheric boundary layer, facing inflow conditions in terms of wind intensity and turbulence that have never been encountered by wind turbines to date (Veers et al., 2019). More significantly, installing a wind turbine on a floating platform means that the system is now subject to a combination of loads coming not only from wind, but also from waves and currents (Chen et al., 2020). Floating wind dynamics are highly coupled, depending on aero/hydro loading, controls, and substructure

and mooring design, with a compliant support structure meaning that aerodynamic forces impact hydrodynamic loading and vice versa, and each control maneuver that induce changes on aerodynamic loads necessarily also affecting the global dynamics of the system (Larsen and Hanson, 2007; Vanelli et al., 2022).

Among the implications of the fully coupled, aero-hydro-servo-elastic response of FOWTs, an aspect that is receiving special attention is the potential impact on turbine wake. On the one hand, studies are being carried out to understand to what extent

blade performance can be affected by an interaction with their own wake in the case of massive turbine displacements (Ramos-García et al., 2022); similar events have been demonstrated to be possible, although probably not as frequently as originally supposed (Papi et al., 2024). On the other hand, it is apparent from many studies that the platform motion may indeed affect wake meandering mechanisms (Fontanella et al., 2022; Kleine et al., 2022; Messmer et al., 2024a). This evidence is even prompting some researchers to speculate on how such motions could be turned into potential advantages, such as further

increasing the wake mixing and thus delivering a more energized flow to the downstream turbines in the farm. Advanced control strategies like the Helix and the Pulse mixing have been proposed for the scope, and numerically investigated for both fixed (Frederik et al., 2020) and floating conditions (Berg et al., 2022; van den Berg et al., 2023).

Overall, it is apparent that the study of FOWTs' wakes is still an open question from many perspectives, with the lack of any experimental validation at large scale still representing an obstacle towards a more complete understanding (Xu et al., 2024).

In response to this limitation, the scientific community is investing efforts in empowering research programs able to provide experiments at wind tunnel scale to validate and tune numerical models (Wang et al., 2021). Among such programs, in Task 30 from the International Energy Agency (OC6 project (IEA Wind, 2022)) significant research has been devoted to investigate aerodynamic models (ranging from state-of-the-art Blade Element Momentum (BEM) codes to Computational Fluid Dynamics (CFD)) and evaluate their ability in reproducing the aerodynamics of FOWTs (Bergua et al., 2023; Cioni et al., 2023). The

OC6 project made use of the experiments carried out in the wind tunnel of Politecnico di Milano (Fontanella et al., 2021), in which a scaled turbine has been mounted on a 6 degrees of freedom (6-DOF) robot and subjected to floating-like pitch and surge motions at different frequencies. While good agreement between the project's participants (and the different levels of fidelity) was found when predicting rotor loads and near wake characteristics (Bergua et al., 2023), more uncertainty was noted

in the mid and far wake (Cioni et al., 2023), with a larger spread between simulations and a large gap from experiments.

Among the hypothesized reasons for such discrepancies, the simplified modeling of inflow turbulence is seen as one of most critical. Indeed, most CFD-based models featured very low turbulence at the rotor plane, while the ones based on lifting-line free vortex wake methods neglected these effects completely (Cioni et al., 2023). While this choice was reasonable in the perspective of a fair comparison between all codes, suggestions are being made about the fact that turbulence may in fact change some of the phenomena described so far in numerical studies. For example, Xu et al. (Xu et al., 2024) studied the impact of a realistic atmospheric flow on a FOWT, showing that the power and thrust had greater instability compared to uniform inflow and shear inflows. Moreover, the atmospheric inflow induced wake breakdown and wake meandering, resulting in a faster wake recovery. Properly accounting for turbulence is particularly relevant to those using URANS CFD (e.g., (Fang et al., 2020), (Arabgolarcheh et al., 2022)), in which Reynolds decomposition leads to instantaneous quantities being decomposed into their time-averaged and fluctuating quantities. The fluctuating velocity field is replaced with a Reynolds stress term, treating the effect of turbulence basically as an additional viscosity. In this regard, properly setting turbulence parameters like turbulent kinetic energy and dissipation rate is critical, as they decay fast along the domain for numerical reasons. Only recently higher fidelity tools have been applied to FOWT wake analysis, with near and far wakes of floating systems being solved with an Actuator Line Method (ALM) combined with LES for the solution of the flow field (Combette, 2023; Fang et al., 2020; Firpo et al., 2024; Yu et al., 2023), although the investigation of the effects of turbulent inflows still remains an open issue. In addition, recent experiments have underlined the influence of turbulence on FOWT wakes. Messmer et al. (Messmer et al., 2024a) performed wind tunnel experiments on a small-scale rotor model and noted improved mixing and wake recovery downstream the moving rotor in laminar conditions. When subject to varying levels of inflow turbulence however, the effects of rotor motion appear to be greatly diminished (Messmer et al., 2024b).

Moving from this background, the present study aims to contribute to the understanding of FOWT wake development, particularly regarding how rotor motion interacts with inflow turbulence. To this end, high-fidelity CFD simulations are used. The computational approach is based on a Large Eddy Simulation (LES) for the flow field, while the turbine is modeled with an Actuator Line Model. Simulations are validated with a new set of experiments, carried out again in the wind tunnel of Politecnico di Milano within the framework of the NETTUNO research project ("*Understanding turbine-wake interaction in floating wind farms through experiments and multi-fidelity simulations*") (NETTUNO, 2023). The turbulence spectrum used in simulations closely reproduces the one measured in experiments. Turbulent simulations are compared to idealized ones with laminar inflow to point out how the macrostructures in the wake of a FOWT (both still and in pitch motion) are affected by turbulence. As an ancillary outcome of the study, analyses are provided to quantify to what extent less computationally expensive wake solution methods such as Unsteady Reynolds-Averaged Navier Stokes (URANS) methods (currently more affordable at industrial level than LES) can be used to reliably study the problem of FOWT wakes. To this end, two URANS set-ups are compared; in the first, which actually represents the standard approach for URANS simulations, inlet turbulence is modelled by imposing appropriate boundary conditions for turbulence parameters (namely turbulence intensity and

lenghtscale), while in the second inflow turbulence is modeled similar to higher-fidelity methods, i.e., by imposing free-stream velocity fluctuations along the boundary.

The work is structured as follows. The case study of the NETTUNO project is first introduced (Section 2). A complete description of the numerical set-up, including the domain, the mesh and the numerical setup necessary to achieve satisfactory accuracy of the results is then provided (Section 3). The results (Section 4) first present phenomenological description of the effects of turbulence on FOWT, comparing the results from LES simulations and experiments, and analyzing the modifications in the main macro structures of the wake. Then, results from the URANS approaches are compared with the LES results, assessing the capabilities of both for similar studies. Finally, Section 5 presents some conclusions and provides handouts for future developments.

## 2  Case study

Numerical analyses have run in parallel and share the test-case with the first stage of the NETTUNO project, a new experimental campaign realised by Politecnico di Milano (PoliMi), which is part of a more extensive work investigating the influence of the wake of a floating wind turbine, on the response of a second downstream turbine. The 1:75 scaled DTU 10 MW wind turbine model was tested within the Galleria del Vento of PoliMi (GVPM). The wind tunnel has an extension of 35 m x 13.84 m x 3.84 m. The turbine is shown in Figure 1, and its main characteristics are reported in the Tab. 1. Further details regarding the experimental test case can also be found in (Fontanella et al., 2021).

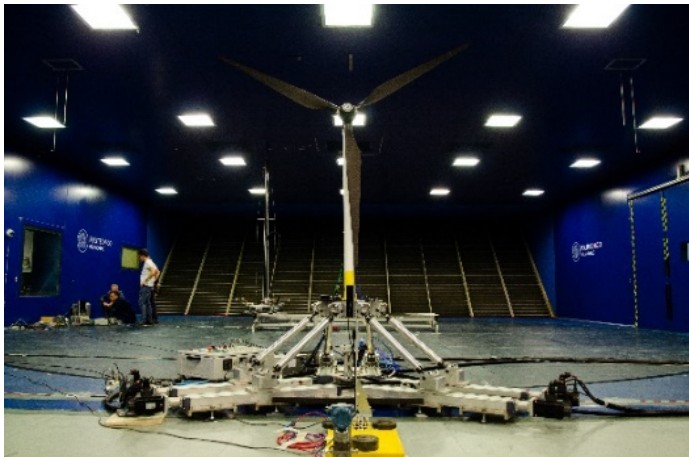 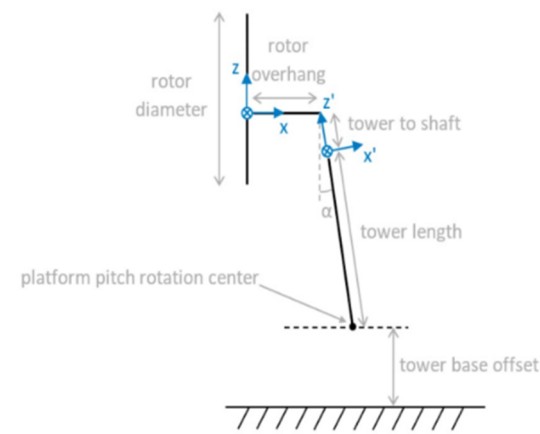

**Figure 1:** Experimental testcase and geometric properties of the scaled DTU 10 MW model.

**Table 1:** Geometric properties of the scaled DTU 10 MW model.

| Parameter | Value |
| --- | --- |
| Rotor diameter [m] | 2.38 |
| Blade length [m] | 1.10 |
| Tilt angle [°] | 0 |
| Rotor overhang [m] | 0.139 |
| Tower-to-shaft [m] | 0.064 |
| Tower length [m] | 1.4 |
| Tower base offset [m] | 0.73 |

120

The wind turbine model was installed on a six-DOF robot that can generate both translational and rotational motions with user-defined frequencies and amplitudes. The wind tunnel was operated maintaining a constant undisturbed wind speed of 4 m/s. For this velocity, the Reynolds number along the blade ranges between $8*10^4$ and $1*10^5$ for most of the blade span (from 30% to 90%) (Fontanella et al., 2021). Turbulence intensity at the rotor is approximately 1.5%, and the air density (constantly measured during the tests) is 1.18 kg/m$^3$ (the temperature inside the tunnel was kept in the range 20 °C ±1). Two cases were considered in the numerical campaign, with the rotor being fixed and in pitching motion, following a fixed sinusoidal law with 1.3° of oscillation amplitude and a frequency of 1 Hz. More details are shown in Tab. 2. The Strouhal number is defined as $St=f_pD/U_\infty$, where $f_p$ is the platform motion frequency, $D$ is the rotor diameter and $U_\infty$ is the undisturbed wind speed. If scaled based on the Strouhal number, the imposed oscillation would result in a 1.3° pitch oscillation with a 25 s period at full scale. This corresponds to a realistic oscillation at a frequency which is representative of the design natural frequency of many FOWT platforms (Behrens de Luna et al., 2024).

In the experiments, axial velocity was measured on two horizontal traverses downstream the rotor plane at a distance of 3 and 5 diameters, respectively. For each traverse velocity was measured in 35 equally spaced acquisition points, from -1.7 m to +1.7 m with respect to the rotor center.

**Table 2:** Main characteristics of the simulation runs.

| Parameter | fixed case | pitching case |
| --- | --- | --- |
| Inflow speed | 4 m/s | 4 m/s |
| Rotor speed / frequency | 240 rpm / 4 Hz | 240 rpm / 4 Hz |
| Blade pitch | 0° | 0° |
| Pitch amplitude | n/a | 1.3° |
| Pitching frequency | n/a | 1 Hz |
| Strouhal number | 0 | 0.595 |
| Tip Speed Ratio | 7.5 | 7.5 |
| Pitching frequency / motion frequency | n/a | 4 |

## 3 Numerical setup

Two different numerical set-ups have been developed for the URANS and LES simulations, respectively, in order to tailor the modelling strategy to the wake resolution method. Aspects such as mesh setup, turbulence model, and near wall modelling have been treated according to each model's requirements. In addition, the URANS numerical set-up was refined based on the free-stream turbulence imposition method that was adopted. In the following of the manuscript, we refer to "URANS" simulations when inflow turbulence is imposed by means of appropriate boundary conditions for the turbulence transport equations, and "URANS-STG" when inflow turbulence is imposed by means of free-stream velocity fluctuations, generated through a synthetic turbulence generator (STG). Simulations have been run with the ALM feature within the software CONVERGE by Convergent Science (Richards et al., 2024). This simulation environment, ubiquitously known for its potential in case of combustion processes and confined flows, indeed provides very interesting features that make it attractive also in turbomachines operating in external flows. In particular, the software allows for autonomous meshing, i.e., it automatically creates the mesh at runtime, dynamically adapts the mesh throughout the simulation. This is a key factor in reducing the operator-dependence of the results. The ALM method has been developed since 2021 in partnership with the University of Florence (Papi et al., 2021) and has been recently applied with success to the simulation of wind turbines (bottom-fixed and floating) in IEA Tasks 47 and Task 30 (OC6) (Cioni et al., 2023).

### 3.1 CFD solver settings

The CFD domain was set to represent the geometry of the wind tunnel testcase and corresponds to the wind tunnel size (Fig. 2a). The robot, tower, nose, and nacelle (Fig. 2b) are included inside the wind tunnel domain. The wind tunnel inlet is placed 8.8 rotor diameters upstream of the rotor, while the domain extends up to 6 diameters downstream. These boundary placements were defined based on the actual wind tunnel test section, which extends a corresponding amount upstream and downstream the rotor, where the turning vanes of the closed circuit do start. The lateral walls of the tunnel are placed 2.8 diameters away from the rotor center, again based on the physical dimensions of the wind tunnel. This is a meaningful element of novelty with respect to past similar studies, as the geometry of the nose cone, nacelle, tower, and robot are often neglected (Arabgolarcheh et al., 2022; Combette, 2023; Fang et al., 2020; Firpo et al., 2024) while, as discussed in section 3.1, these components interact in a non-negligible way with the wake of the wind turbine. The blades are modelled using the ALM approach, explained in detail in section 3.2. Dirichlet boundary conditions for velocity, temperature, and sub-grid turbulent kinetic energy are imposed at inlet for the LES simulations, while turbulence intensity and length scale are imposed for the URANS simulations. In this second case, turbulence parameters are fine-tuned in order to compensate for the numerical decay along the computational domain, so that the actual length scale and turbulence at the rotor are consistent with the experiments. In the LES and URANS-STG simulations, turbulence is accounted for by inserting velocity perturbations inside the flow field, as detailed in section 3.3. A Dirichlet boundary condition is also imposed at the outlet for pressure, while a Neumann boundary condition is imposed for velocity, temperature and sub-grid turbulent kinetic energy. The geometries of ground, robot, tower, nose cone, and nacelle

are set as no-slip walls, while lateral and upper walls are set as free-slip to avoid explicitly solving the boundary layer. In line with this choice, and similarly to what was done during the OC6 project (Bergua et al., 2023; Cioni et al., 2023), these walls have been moved slightly inward the domain to account for boundary layer blockage on the free-stream flow. Regarding turbulence closure, the Realizable k-ε model was chosen for the URANS simulations, with standard wall functions for the near wall treatment. For LES simulations, the One-Equation Viscosity Model, as formulated by (Menon et al., 1996; Yoshizawa and Horiuti, 1985), was adopted, with the treatment of velocity profiles close to the walls (the so-called "law of the wall") resolved according to the Werner and Wengle model, which best fits for internal flows and has been validated by simulating the flow around a cube on a plate channel (Werner and Wengle, 1993).

## 3.2 Mesh sensitivity

The mesh setup adopted for the URANS simulations is shown and presented in Appendix B. A preliminary grid sensitivity study was performed in the rotor wake to ensure that wake characteristics are insensitive to the numerical resolution. Mesh refinement in the rotor region was not varied during such sensitivity analysis because the mesh size had been already tuned in conjunction with the ALM set-up during the OC6 project (Bergua et al., 2023; Cioni et al., 2023). Insensitivity of the wake profile was tested in terms of both convergence history in time and accuracy in comparison to experimental data, as shown in Fig. 3. This sensitivity analysis - performed with a 2% turbulence intensity at the inlet in accordance with preliminary estimations - led to a mesh of approximately $15*10^6$ elements, indicated as M3 and described in Tab. 3. It has to be remembered that CONVERGE features an Automatic Mesh Refinement (AMR) feature, which allows for selective refinement of the computational grid based on gradients in field variables (Papi et al., 2021). For the simulations presented in this work, the criterion adopted for the AMR is based on the velocity field, so that the mesh resolution is automatically enhanced where the gradients of velocity are higher than the user-specified threshold, which was fine-tuned during preliminary test-runs to ensure additional cells were added to critical regions such as the inner and outer shear layers of the wake, as shown in Fig. B1.

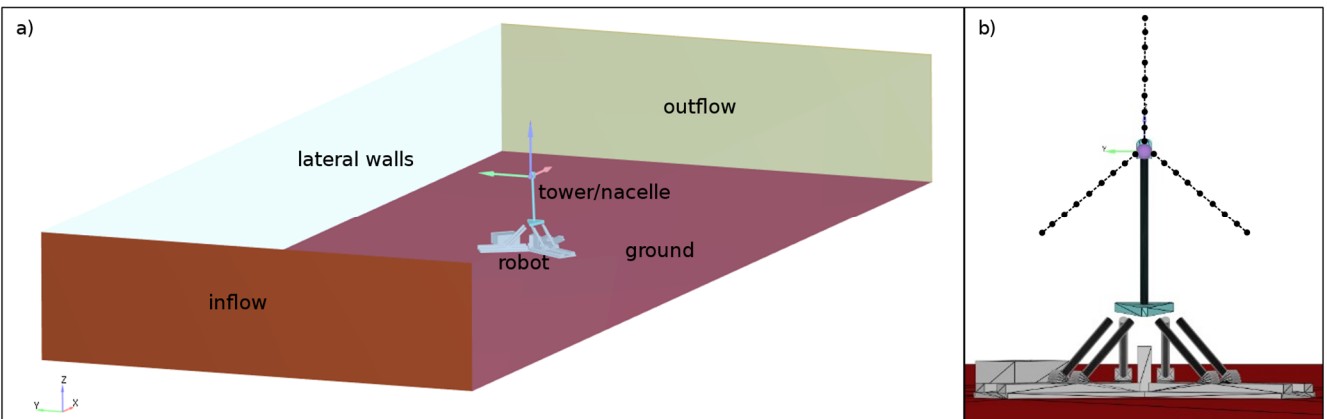

**Figure 2:** CFD domain: complete geometry (a) and focus on robot, tower and nacelle geometries (b); rotor represented with fictitious ALM lines (number of nodes not consistent with the setup).

In this case, as demonstrated by $R^2$ values in Tab. 3, the reduction in the AMR activation threshold (Tab. 3) in M2 brought mean wake velocity in closer alignment with experimental data (Fig. 3 (a)). On the other hand, a much smaller improvement

can be noticed when M2 and M3 are compared, which differ mostly in the vertical traverse (Fig. 3 (b)). These differences, however, are mostly located in the bottom shear layer of the wake, which, as explained in the following sections, is heavily affected by the robot's wake. Despite minor differences with respect to M2, which would suggest this set-up to be adequate, M3 was ultimately chosen. This was a precautionary choice and is justified by the desire to accurately solve wake structures also in the pitching case. For the URANS-STG simulation discussed in chapter 4.3, a similar mesh is adopted, except for the

box embedding region required for the turbulence injection in front of the rotor, which is similar to the LES mesh setup. Moreover, AMR was deactivated, and a fixed embedding was set to obtain the same discretization of the wake.

**Table 3:** URANS meshes tested during the sensitivity analysis. The coefficients of determination $R^2$ are computed between each mesh and the experiments for the horizontal traverse at 3D from the rotor shown in Fig. 4 (a).

|  | M1 | M2 | M3 |
|---|---|---|---|
| Mesh size around the robot | 0.013D | 0.0065D | 0.0065D |
| AMR threshold | 0.01 | 0.005 | 0.0025 |
| Maximum mesh elements | 6.4M | 11.5M | 15M |
| End of near wake embedding | 0.8D | 0.8D | 3D |
| $R^2$ @3D with respect to EXP | 0.334 | 0.643 | 0.664 |

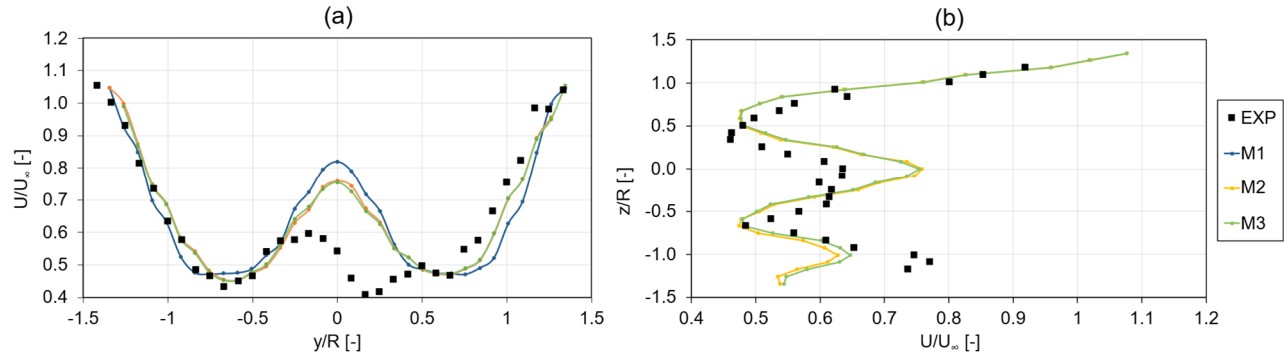

**Figure 3:** Mesh and wake convergence; horizontal and vertical traverses at 3D from the rotor.

LES simulations required a finer grid than their URANS counterparts. The mesh setup is shown and discussed in Appendix B and results in an element count of approximately $120*10^6$ elements, nearly 8 times the element number of the URANS setup, resulting in a not negligible increase of computational cost. On a cluster with AMD EPYC 7413 24-Core processors and

210 approximately 200 GB RAM for each node, a single revolution run with the URANS approach requires a maximum of roughly 320 core hours per revolution on two nodes, while a LES simulation requires 2800 to 5000 core hours for the same revolution on a maximum of ten nodes, depending on the complexity of the testcase. It must also be considered that the number of

revolutions required for convergence is not the same for all the simulations, as more time is required for the "statistical convergence" of turbulent LES simulations, further discussed in section 3.4.

### 3.3 ALM blade model

The rotor blades are modelled with an ALM tool, implemented inside CONVERGE. This method was first proposed from a theoretical point of view by (Sørensen and Shen, 2002) and it consists in replacing the physical geometry of the blade with dynamically equivalent momentum sources, evaluated through tabulated polar data given as input to the model. The other variables at hand are the velocity of the flow domain, for the calculation of the angle of attack, the Reynolds number, which is sampled from the CFD domain through a sampling algorithm, and the properties of the distribution function adopted for the spreading of the momentum sources back into the CFD domain, namely the Regularization Kernel (RK). Both velocity sampling and force insertion are non-trivial issues. The former has been here addressed through the line average approach (Jost et al., 2018; Melani et al., 2020), whereby the inflow velocity is obtained by averaging the inflow velocity over a circle at one chord around the ALM point.

Force insertion is made using a piecewise smearing function (Xie, 2021). While similar in shape to the more commonly used Gaussian function, the former has the benefit of being conservative in formulation. The setup of the Regularization Kernel is particularly important for avoiding instabilities in the CFD code, as the insertion of point forces inside the domain would lead to instability issues. On the other hand, spreading the forces to a large volume would lead to a dispersion of the rotor effects, underestimating the induction on the incoming inflow. Therefore, the width of the Regularization Kernel is fine tuned to the blade chord and mesh size. As explained in detail in (Papi et al., 2021; Xie, 2021), the Regularization Kernel width varies along the blade span and is determined as $\beta = \max(c/4, \zeta\Delta)$ where $c$ is the local chord, $\Delta$ is the local mesh size and $\zeta$ is a tuning factor. Different values of $\zeta$ were used in the LES and URANS set-up to ensure consistency in kernel size. In particular, $\Delta = 0.01$ m and $\zeta = 2.4$ in the LES simulations and $\Delta = 0.015625$ m and $\zeta = 1.6$ in the URANS simulations. These values were chosen based on previous validation (Bergua et al., 2023; Cioni et al., 2023). In addition to ensuring consistency between the two simulation approaches, the chosen values ensure $\beta = c/4$ up to approximately 50% of the rotor span, while limiting the kernel size to $\zeta\Delta$ in the outer part of the blade to improve numerical stability.

The blade is discretized with 55 uniformly distributed sections. This choice, together with the RK calibration and the Line Average velocity sampling strategy, is the same used by the authors during the OC6 project (Bergua et al., 2023; Cioni et al., 2023). This choice ensured that the final rotor response simulated with the URANS approach is aligned with most of the other numerical codes in the OC6 project, as well as the experimental results.

As widely demonstrated (Dağ and Sørensen, 2020; Hodgson et al., 2022; Martínez-Tossas et al., 2018), the accuracy in reconstructing tip effects in horizontal-axis wind turbine (HAWT) blades is key for the reliability of the integral rotor response. CONVERGE can use the Dağ-Sørensen model (Dağ and Sørensen, 2020), recently used by some of the authors in the case of fixed wings and vertical-axis wind turbines, and the model proposed by Meyer Forsting et. (Meyer Forsting et al., 2019). It is a computationally efficient model based on a Lifting Line Theory approach for the correction of the angle of attack, with the

scope of accounting for the downwash effects dissipated by the over-diffusion proper of the ALM forces smearing in the near tip region. This latter method was used in the present study.

### 3.4 Turbulence insertion in scale-resolving simulations

In LES and URANS-STG simulations, rather than imposing fluctuations in flow velocity at the inlet, they are injected as volume forces close to the rotor to avoid numerical diffusion from the coarse mesh in front of the rotor. To this end, a distribution of momentum forces in front of the rotor was defined, with the module of each source defined as in Eq. 1 for each Cartesian component (here, the only $x$ component is shown as an example):

$$f_x = C(x)\rho\big[(U_{x;\infty} - U_x) + \Delta u\big] \tag{1}$$

where $\Delta u$ is the desired velocity perturbation in the flow downstream the insertion zone, $U_{x;\infty}$ is the undisturbed inlet velocity and $U_x$ is the local velocity at the insertion zone, $\rho$ is the air density and $C(x)$ is an exponential distribution function, which distributes the forces axially in the insertion zone. Additional details are shown in Appendix A. The desired perturbation $\Delta u$ is obtained by subtracting the mean undisturbed velocity from the desired turbulent inflow velocity $\Delta u = U - U_{x;\infty}$. The turbulent inflow velocity was generated from the turbulent spectrum measured in the wind tunnel (shown in Appendix A), with a maximum frequency of 100 Hz. In fact, very low energy was noted in the turbulent spectrum beyond this frequency. Therefore, an insertion frequency of 200 Hz was chosen, according to the minimum requirements in terms of signal reconstruction. From the turbulent spectrum a synthetic turbulent wind field was obtained using TurbSim (Jonkman, 2016) with IEC Kaimal spatial coherence model. The generation zone extends along the entire test section in the vertical direction, and for 5 m (≈ 2D) in the horizontal direction, to ensure the rotor and wake are completely immersed in the free-stream turbulence. This approach was inspired by the work of (Gilling and Sørensen, 2011), where an actuator disk approach is proposed for the turbulence injection. From this starting point, similarly to what proposed in (Spyropoulos, 2024), the planar distribution of momentum sources has been extended in the axial direction through the definition of an exponential distribution function. The input velocity perturbations were calculated directly from the spectrum derived from the experimental velocity sampling and conveniently scaled to compensate for numerical dissipation in the insertion process. The choice of the insertion plane, both in terms of dimensions, spatial and temporal discretization, and distance from the rotor is key to obtaining the desired turbulent characteristics at the rotor plane. Some parameters have been found to be strongly related to each other, such as temporal and spatial discretization, which must be appropriately related to the turbulence spectrum characteristics and to the rotor size. Conversely, others have been found to be more set-up dependent. For instance, the location of the insertion plane is suggested at 1R upstream the rotor in (Spyropoulos, 2024), while in IEA Task 29 – Phase IV project (Schepers et al., 2021, p.29), turbulent fluctuations are inserted at 320m (≈ 4D) distance from the NM80 turbine. In the present study, based on dedicated tests, the generation zone has been positioned at 2 diameters in front of the rotor. More details on the obtained turbulent characteristics at the rotor plane are shown in Appendix A.

### 3.5 Simulation length and statistical convergence in scale-resolving simulations

Convergence was checked for main rotor performance metrics and wake profile. Once full convergence was reached, the simulation time recorded for data analysis was 5 s, corresponding to 5 pitching cycles in the pitching simulations and 20 rotor revolutions. This ensured that the variation of torque with respect to the previous revolution was less than 1% in all cases. The only exception was represented by turbulent simulations using LES and pitching rotor, in which the varying inflow, coupled with platform motion, caused slow statistical convergence of mean wake values. In this case, a 10 s sampling window, corresponding to 10 pitch cycles, was necessary to ensure proper convergence at 5D downstream the rotor plane, as shown in Tab. 4 and Fig. 4. In Fig. 4 the first two statistical moments, i.e. the mean and the standard deviation of the axial velocity sampled at 3D and 5D are shown. For these two moments, which form the basis of the subsequent analyses in this study, good convergence can be noted. Selected sampling intervals are in good agreement with the suggestions provided by (Martínez-Tossas et al., 2018). All numerical results are averaged over multiple rotor revolutions. Finally, due to the intrinsic stochasticity of the scale-resolving simulations, when relevant the cyclic results are phase-averaged during one pitching cycle, in order to obtain the mean response cycle for the quantity of interest.

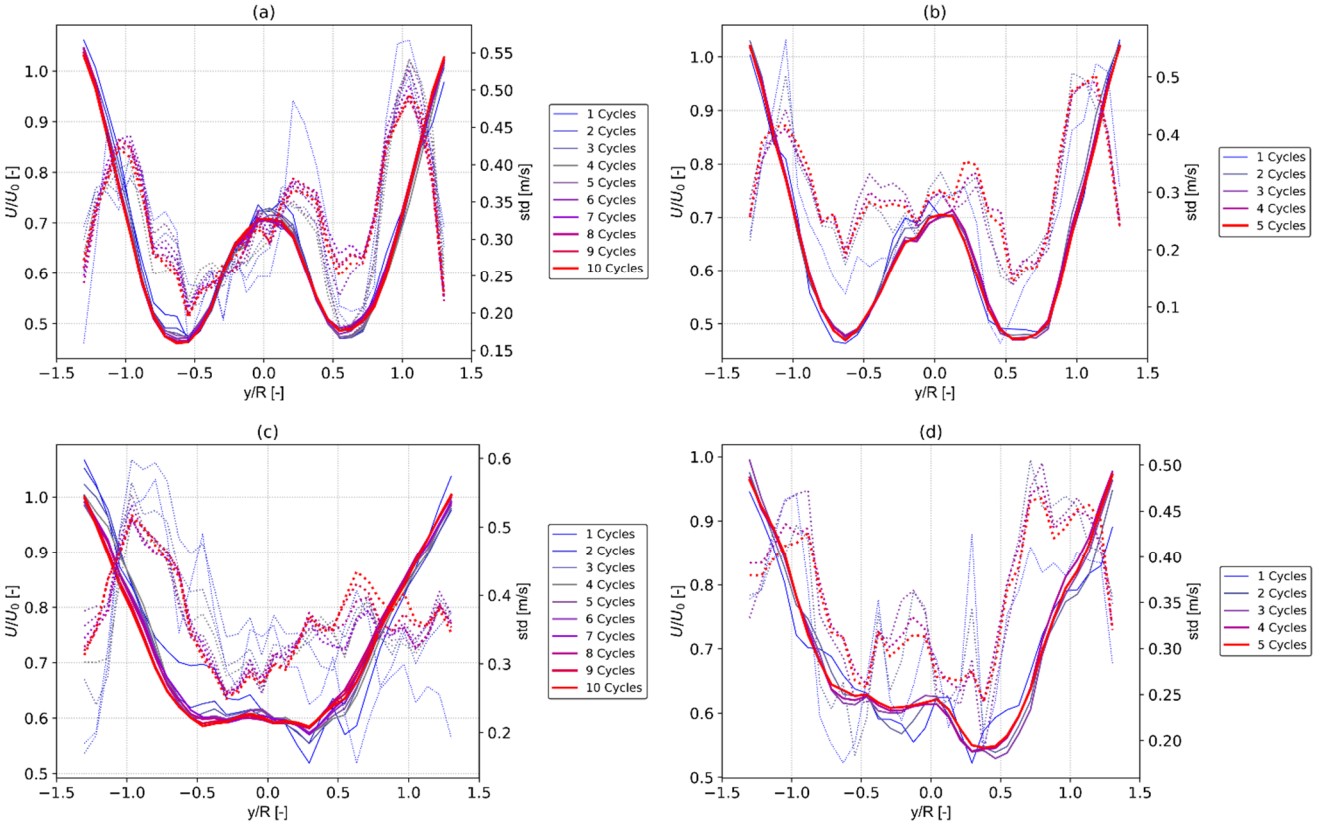

**Figure 4:** Mean velocity profiles (solid lines – left axis) and standard deviations (dashed lines – right axis) for LES_t cases at (a) 3D – pitching case, (b) 3D – fixed case, (c) 5D – pitching case, (d) 5D – fixed case.

**Table 4:** Coefficients of determination $R^2$ for each number of pitching cycles taken for averaging, with reference to the wake averaged over the respective maximum number of cycles.

| n° of cycles | 3D - pitching | 3D - fixed | 5D - pitching | 5D - fixed |
|:---:|:---:|:---:|:---:|:---:|
| 1 | 0.960 | 0.984 | 0.875 | 0.898 |
| 2 | 0.985 | 0.989 | 0.952 | 0.973 |
| 3 | 0.992 | 0.997 | 0.963 | 0.989 |
| 4 | 0.992 | 0.997 | 0.969 | 0.996 |
| 5 | 0.995 | 1.000 | 0.986 | 1.000 |
| 6 | 0.996 | | 0.992 | |
| 7 | 0.997 | | 0.989 | |
| 8 | 0.998 | | 0.992 | |
| 9 | 0.999 | | 0.999 | |
| 10 | 1.000 | | 1.000 | |

## 4 Results

Simulations analyzed in this section consist of eight cases, characterized by a combination of the key metrics investigated in this study, i.e., fixed or pitching platform, laminar or turbulent inflow, URANS or LES solution of the flow field. For the sake of brevity, simulations will be referred to as follows throughout this section:

- **URANS_l** – URANS simulation with laminar inflow
- **LES_l** – LES simulation with laminar inflow
- **URANS_t** – URANS simulation with turbulent inflow
- **LES_t** – LES simulation with turbulent inflow
- **URANS_stg** – URANS simulation with turbulent inflow ("h" stands for "hybrid", as indicated in Subchapter 3.1)

### 4.1 Rotor performance

Good agreement can be generally noted for mean rotor quantities among all setups (Tab. 5) whenever no platform motion is included in the analysis, with numerical results highlighting higher torque and lower thrust than the experimental results. According to (Fontanella et al., 2024), possible causes for these deviations are small differences in inflow velocity in the wind tunnel and small blade pitch errors in the experimental model. Importantly for this study, all four numerical models are in close alignment with each other. Mean rotor forces, predicted by numerical models or measured in experiments with pitching motion, are shown in Tab. 6. For this case, details about the cyclic variations of thrust and torque during the pitch cycle are included. Similar to the fixed case, all numerical approaches are quite well aligned, with relative differences below 1% for all quantities except for the URANS torque values.

**Table 5:** Rotor thrust and torque for the fixed case and relative difference in thrust (ΔTh) and in torque (ΔTq) with respect to experimental results (EXP); for experimental results, the maximum and minimum variation in mean values recorded during different runs is reported.

| | Torque [Nm] | ΔTq [%] | Thrust [N] | ΔTh [%] |
|---|---|---|---|---|
| EXP | 2.953 | +6.23<br>-11.94 | 36.469 | +2.61<br>-4.73 |
| LES_t | 3.241 | +8.9% | 35.296 | -3.3% |
| LES_l | 3.137 | +5.9% | 34.877 | -4.6% |
| URANS_t | 3.211 | +8.0% | 35.184 | -3.7% |
| URANS_l | 3.203 | +7.8% | 35.149 | -3.8% |

**Table 6:** Pitching case: mean values and maximum and minimum deviations from the mean value, for the phase-averaged variations of rotor thrust and torque; relative difference of the mean values of thrust (ΔTh) and in torque (ΔTq) from the LES simulation with turbulent inflow (LES_t).

| | Torque [Nm] | ΔTq [%] | Thrust [N] | ΔTh [%] |
|---|---|---|---|---|
| LES_t | $3.169^{+0.473}_{-0.469}$ | +0.0% | $34.988^{+1.916}_{-1.946}$ | +0.0% |
| LES_l | $3.143^{+0.479}_{-0.472}$ | -0.8% | $34.881^{+1.947}_{-1.965}$ | -0.2% |
| URANS_t | $3.194^{+0.483}_{-0.475}$ | +0.8% | $35.042^{+1.932}_{-1.950}$ | +0.2% |
| URANS_l | $3.217^{+0.487}_{-0.478}$ | +1.5% | $35.173^{+1.963}_{-1.965}$ | +0.5% |

## 4.2 Effects of turbulence of FOWT wake

This section analyses the results from LES simulations, assumed as the closest numerical reproduction of reality, to assess the effects of turbulence on the wake of FOWTs. Integral quantities, mean spanwise values, and contours of the most representative flow variables are shown, trying to highlight the most interesting differences between fixed and pitching platform cases, either with laminar or turbulent inflow.

Comparing first normalized streamwise velocity in Fig. 5 (a), sound match between experiments and numerical results can be noted, especially for the LES_t case, which closely matches the right-hand shear layer, and maintains high accuracy on the left one. When the LES_t and LES_l cases are compared, the inclusion of free-stream turbulence apparently improves the agreement in normalized mean velocity 3D downstream the rotor with respect to experiments in the outer shear layers. As shown in Fig. 5 (a), LES_l results are similar to those of Firpo et al. (Firpo et al., 2024), which were obtained with a similar LES-ALM model. The main difference between the previous study is in the center of the wake, and it can be ascribed to the lack of the tower and nacelle in the study by (Firpo et al., 2024). A discrepancy with respect to experiments is still noticed in the center area of the wake for both the LES_l and LES_t cases, where a different degree of symmetry clearly stands out. A possible explanation could be a discrepancy in spanwise force distribution at blade root between experiments and numerical models; this hypothesis is still under investigation by the authors. The inclusion of inflow turbulence in the LES_t case clearly

reflects also in a change of turbulence intensity at 3D downstream, as can be observed in the plots of streamwise turbulence intensity (Fig. 5 (b)). Turbulence intensity is computed as in Eq. 2:

$$I_{x;LES} \approx I_{x;res} = \frac{u'_x}{\overline{U}}. \tag{2}$$

As shown in Eq. (2), the resolved turbulence intensity is computed by normalizing the standard deviation of the velocity fluctuations $u'_x = std(U - \overline{U})$ by the mean velocity $\overline{U}$. When free-stream turbulence is considered, a noticeable improvement – in terms of agreement with experiments – is reached in the shear region, where the addition of approximately 1.5% free-stream TI causes an increase of 10-15% in turbulence intensity. Such an increase is due to the earlier wake transition that turbulence promotes. While present, this effect is less pronounced in the central part of the wake, presumably due to the fact

that the closely-spaced root vortices and turbulence generated by the hub and the nacelle already promote mixing in the laminar simulations.

It is also worth noting that the shape of the wake at 3D from the turbine for the laminar inflow case (Fig. 5 (a)), greatly resembles previous numerical campaigns (Bergua et al., 2023). This is to be expected as inflow turbulence played a minor role in such numerical endeavors and was mostly neglected by those using lifting-line free vortex wake codes.

If the same comparison of mean quantities is performed at 5D from the rotor (Fig. 6), the improved agreement with experiments of the LES_t simulation is still noticeable, especially for the streamwise turbulence (Fig. 6 (b)), but less apparent. Such a difference from 3D to 5D could indicate a different wake evolution trend, especially for negative y/R values.

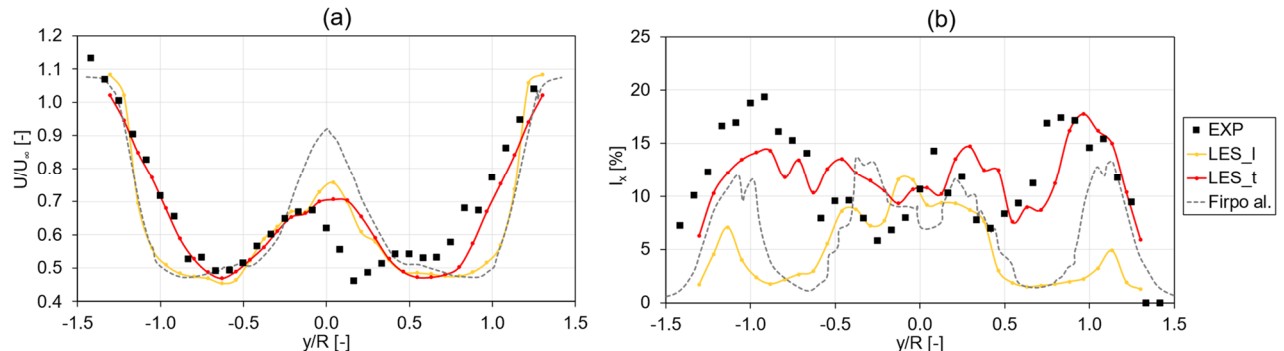

**Figure 5:** Fixed case, wake profiles of velocity at 3D, horizontal (a) and vertical (b) traverse LES simulations with laminar and turbulent inflow conditions; comparisons with experiments (EXP) and results from LES simulations – with laminar inflow and without solid geometries – in literature (Firpo et al., 2024).

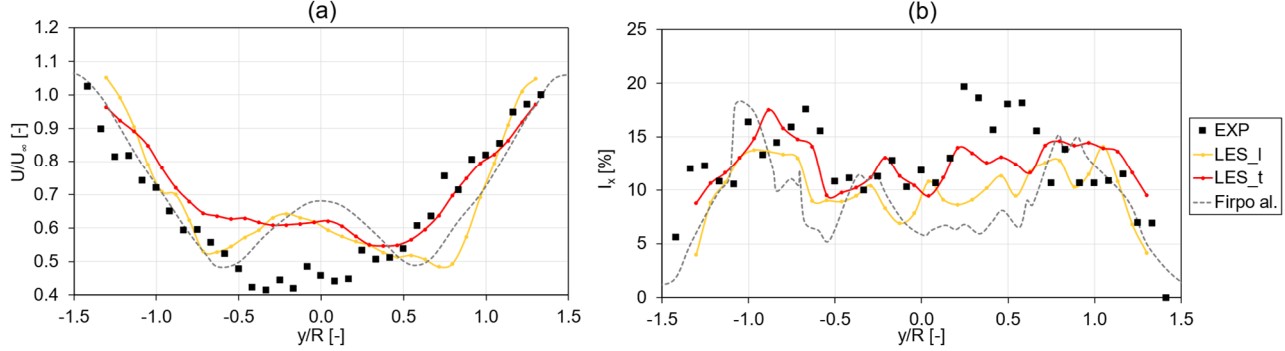

**Figure 6:** Fixed case, wake profiles of velocity at 5D, horizontal (a) and vertical (b) traverse LES simulations with laminar and turbulent inflow conditions; comparisons with experiments (EXP) and results from LES simulations – with laminar inflow and without solid geometries – in literature (Firpo et al., 2024).

The overall effect of including free-stream turbulence on the mean velocity profiles is similar to what was observed at 3D from the rotor, with increased mixing and a larger shear layer when inflow turbulence is included, as shown in Fig. 6 (a). Moreover, the mean velocity profiles obtained with inflow turbulence more closely resemble a single-Gaussian wake deficit profile, which is characteristic of the far wake of a wind turbine, rather than the double-Gaussian deficit that is found in the near-wake. This is the first indication of the fact that the inclusion of inflow turbulence has the effect of accelerating wake dissipation and near to far wake transition. The differences noted in the left part of the wake in Fig. 6 (a) could be due to a different evolution of root vortices, which are incorrectly captured in the simulations even at 3D (Fig. 5 (a)).

When platform pitching is included, similar conclusions can be drawn, but with some notable differences. Starting from the wake profiles at 3D distance, good agreement – in absolute terms – is noted between LES_t and experiments. Similar to fixed cases (Fig 7 (a)), the inclusion of free stream turbulence in the LES_t case still improves the agreement with the experiments, but to a lesser degree. As the wake develops and moves to 5D distance (Fig. 8), LES simulations tend to coalesce to the same trend, and get only marginally closer to experiments (again, similar to what was noted in the fixed case in Fig. 6). Rotor pitching motion seems to have a very similar effect than free-stream turbulence, as it accelerates wake dissipation and moves the near-to-far wake transition point upstream. However, the two effects (i.e., inflow turbulence and rotor pitching) do not combine in a linear manner, as rotor motion does not significantly increase wake recovery when inflow turbulence is considered. Such a non-linear combination between inflow turbulence and platform motion is in agreement with recent observations from similar experimental test cases (Messmer et al., 2024b).

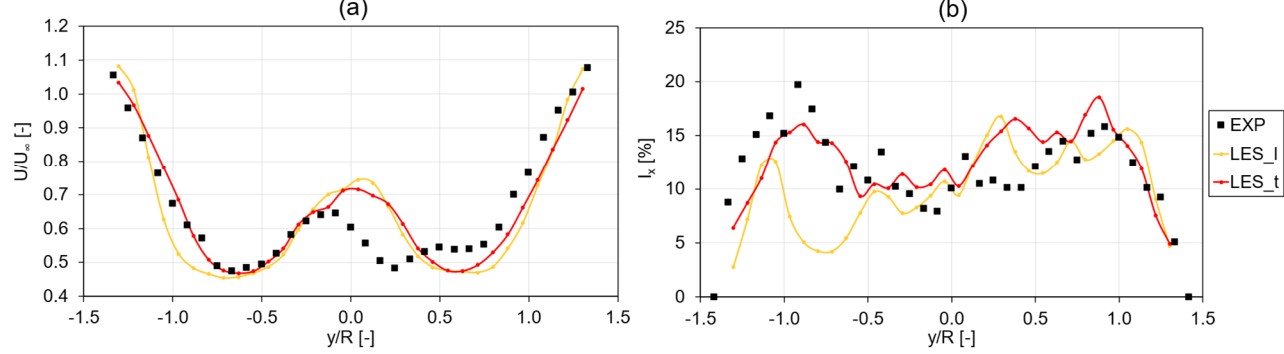

**Figure 7:** Pitching case, wake profiles of velocity at 3D, horizontal (a) and vertical (b) traverse LES simulations with laminar and turbulent inflow conditions; comparisons with experiments (EXP).

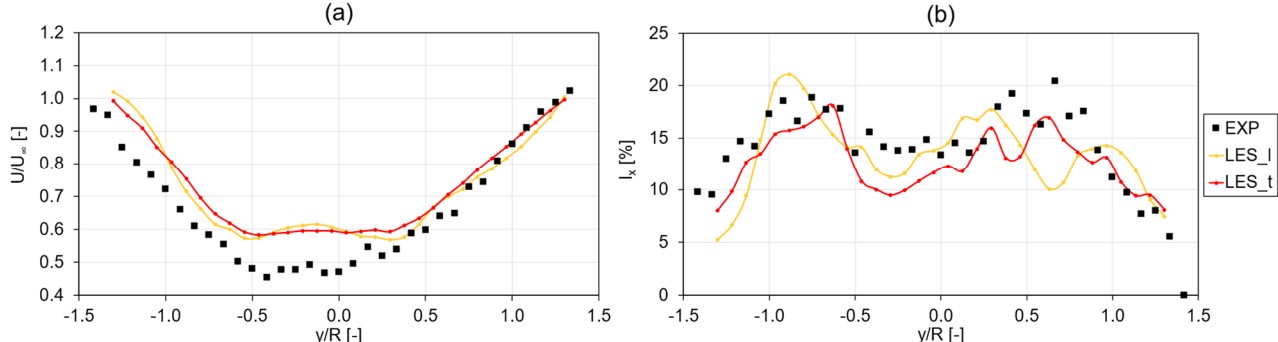

**Figure 8:** Pitching case, wake profiles of velocity at 5D, horizontal (a) and vertical (b) traverse LES simulations with laminar and turbulent inflow conditions; comparisons with experiments (EXP).

To more clearly highlight the reasons underlying the observed differences between the LES_t and LES_l simulations, instantaneous contours of velocity magnitude are shown in Fig. 9, while iso-surfaces of q-criterion are shown in Fig. 10. The effect of inflow turbulence on the wake structures of both the fixed and pitching cases is evident in both figures. In particular, the LES results for the fixed cases (Fig. 9 (a, b), Fig. 10 (a, b)), show that the tip-vortex breakdown and the consequent transition from near to far wake is significantly moved upstream when inflow turbulence is considered. In fact, if we consider the laminar simulation (Fig. 10 (c)), evenly spaced tip vortices can be noted up to approximately 1.8 to 2D downstream the rotor. Some instabilities in these coherent structures can be noted from approximately 2D to 3D, where the wake starts its transition to the chaotic structures that can be seen from 3D downstream. In the turbulent simulations, on the other hand, transition is significantly anticipated (Fig. 10 (b)). Vortex pairing or leapfrogging, whereby two or more tip vortices interact and overlap with each-other, can be noted at approximately 2D downstream the rotor, leading to a much earlier wake breakdown, and a faster transition to the chaotic vortices typical of the far wake of a wind turbine. On the other hand, the inclusion of platform motion in the laminar case (Fig. 9 (c) and Fig. 10 (c)) clearly shows that the wake breakdown, still located

at 3D, is much more abrupt, with the preliminary disturbance at approximately 1.8 to 2D that also seems to be moved inward the wake.

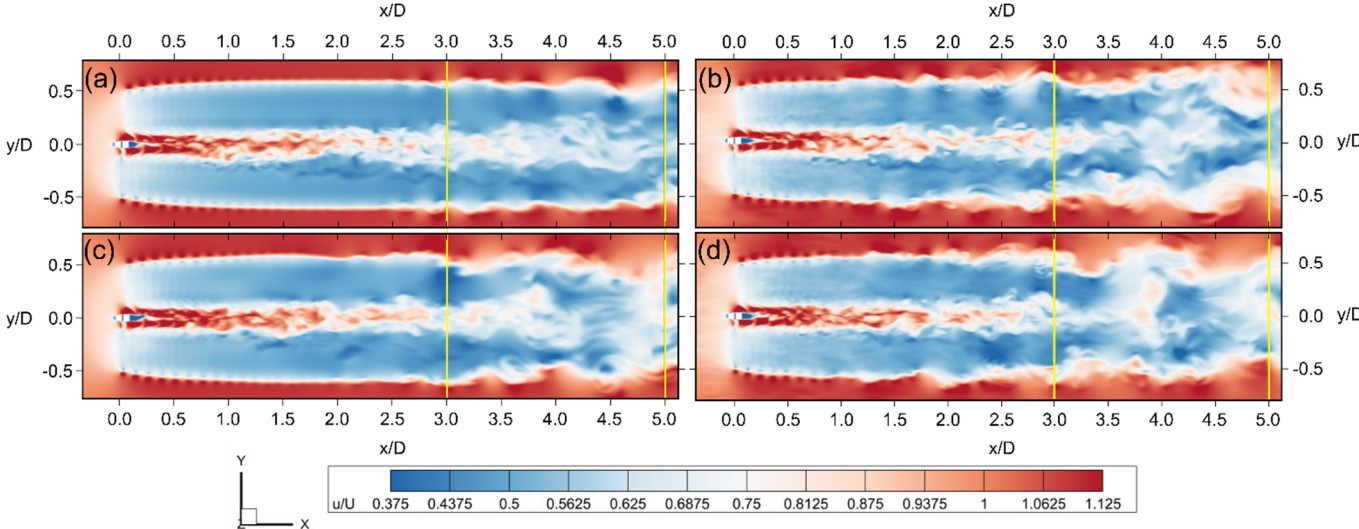

**Figure 9:** LES simulations, velocity contour for the fixed case with laminar (a) and turbulent (b) inflow, and for the pitching case with laminar (c) and turbulent (d) inflow, top view; vertical lines at 3D and 5D. Distances are reported in diameters from the rotor center.

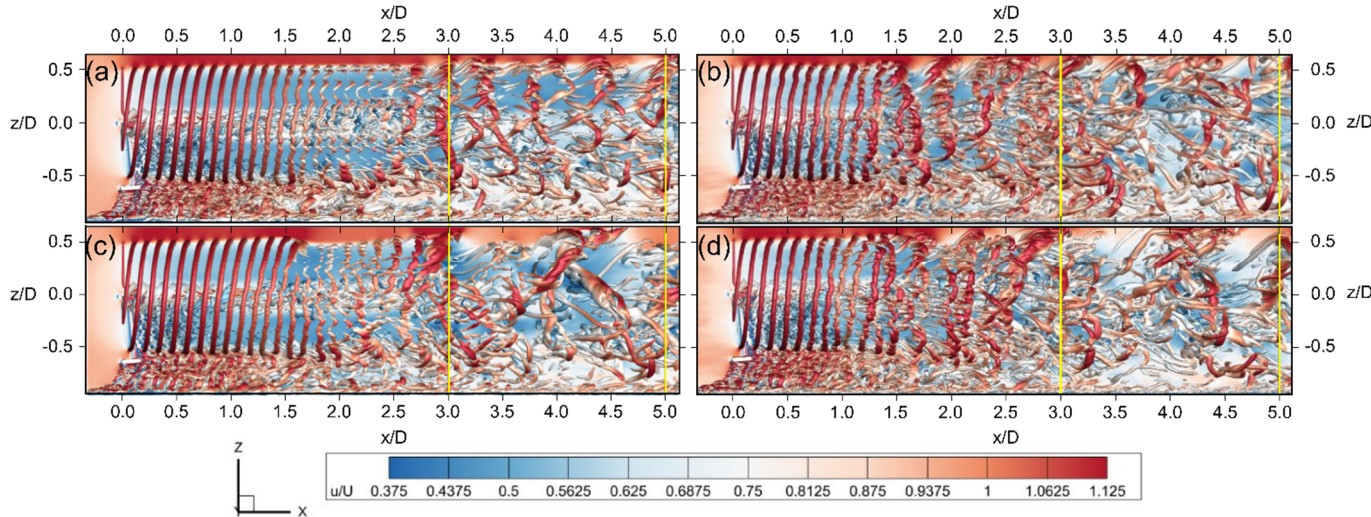

**Figure 10:** LES simulations, velocity contour and iso-surfaces of Q-criterion for the fixed case with laminar (a) and turbulent (b) inflow, and for the pitching case with laminar (c) and turbulent (d) inflow, lateral view; vertical lines at 3D and 5D. Distances are reported in diameters from the rotor center.

If the inflow turbulence is included in the pitching case (Fig. 9 (d) and Fig. 10 (d)), its influence on the behavior of the wake is noticeable even at 1D distance from the rotor, as can be easily noticed by the weakening of the tip vortex structures and the

formation of leapfrogging vortex rings. Moreover, the behavior of the fixed and pitching case with turbulent inflow looks very similar over the first 3 diameters, with some minor differences in the vortex rings that arise in the far wake.

A first conclusion from this contour is that the impact of turbulence on a floating turbine is more relevant in the first diameters of distance; beyond this length, the motion-induced turbulence rules the wake recovery. Further proof of this conclusion can be found in Fig. 11, where the integral values of wake deficit over the entire rotor wake span and at 1-2-2.5-3-5D distance from the rotor are calculated as per Eq. 3:

$$\frac{WD}{U_0} = \frac{\int (U - U_0)dA}{A U_0},$$
(3)

where the integrated area $A$ is the rotor area.

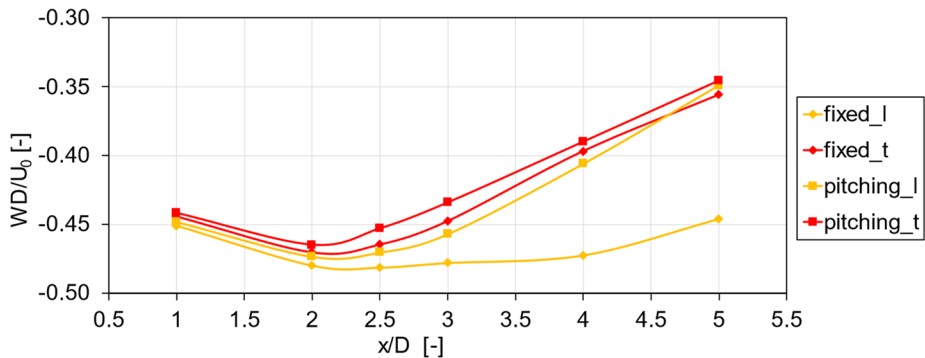

**Figure 11:** Along wake profile of normalized integral Wake Deficit, comparison between results with fixed platform and laminar (F-L) or turbulent inflow (F-T) and the ones with pitching platform and laminar (P-L) or turbulent inflow (P-T).

For all numerical models, axial velocity downstream the rotor reaches a minimum between 2D and 3D and then starts to recover. As expected, the laminar fixed case (fixed_l) shows the slowest recovery, with a roughly 7% velocity increase from its minimum value at 2.5D. It is important to note that, despite the minimum value being reached at 2.5D, axial minimum velocity stays roughly the same at 2.5D and 3D in the fixed_l case, and proper recovery starts from 4D downstream. On the other hand, the point where axial velocity starts increasing and the wake starts recovering is moving significantly upstream when inflow turbulence and rotor motion are included. In fact, the point where minimum velocity is recorded is moved upstream to 2D for both the laminar and turbulent pitching cases (pitching_l/pitching_t) and for the turbulent fixed case (fixed_t). Moreover, while rotor motion significantly increases wake recovery in laminar simulations, the effect is substantially diminished in turbulent cases. Indeed, when turbulent simulations are considered, rotor motion appears to move the wake transition point slightly upstream, but the trend in axial velocity increases downstream of this point is similar. Interestingly, if the pitching_l simulation is compared to turbulent cases, despite reaching a lower minimum speed at 2D, it shows stronger recovery from 3D downstream. This trend can be related to the behavior shown by the wake at 2D and highlighted by Fig. 10 (c): in fact, in absence of free-stream turbulence, the wake breakdown is more abrupt and, although it starts further downstream than in turbulent cases, it shows steeper wake recovery.

Additional insights can be provided by analyzing the spatial distribution of axial velocity, from which integral values shown in Fig. 11 are derived. The difference in mean axial velocity between the pitching and the fixed-bottom cases with or without the inclusion of free-stream turbulence is shown in Fig. 12.

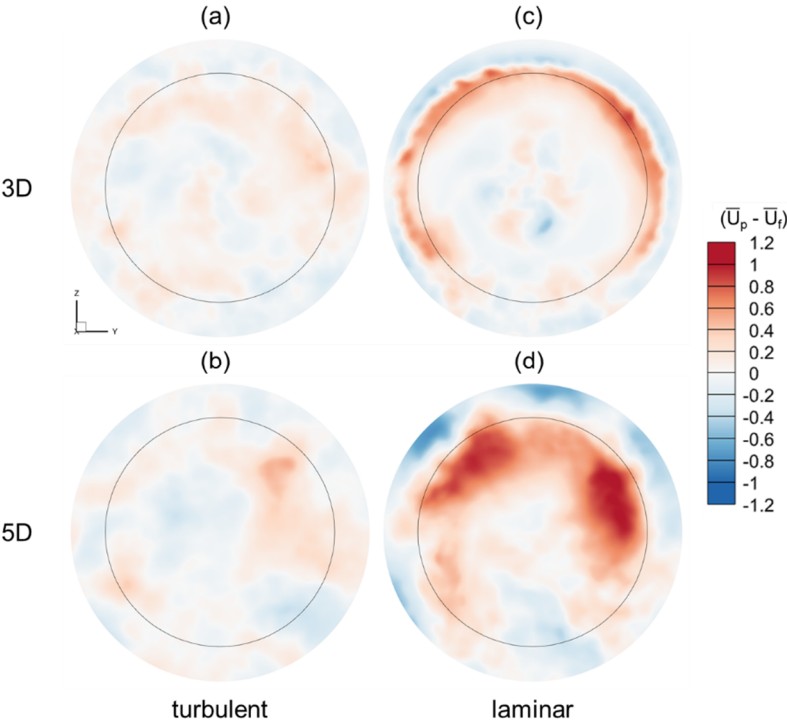

**Figure 12:** LES turbulent simulations, contours of differential wake deficit between the pitching and the fixed case, averaged over the last 5 seconds of the simulations for all the cases except for the pitching turbulent one, which is averaged over the last 6 pitch cycles. The rotor trace is indicated with a black circle.

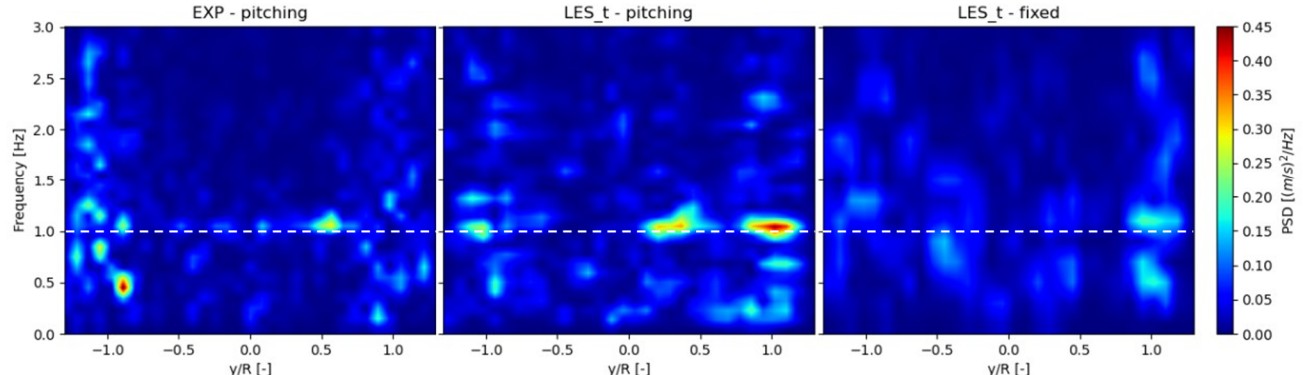

**Figure 13:** Pitching case, contours of PSD calculated from the velocity field sampled at the 3D horizontal traverse; results from experiments (EXP - pitching) and LES with turbulent inflow (LES_t - pitching) compared to fixed case (LES_t – fixed); the motion frequency is highlighted with dashed lines.

In laminar cases, higher velocity for the pitching case can be seen in the shear layer at 3D, and in the upper part of the rotor area at 5D. At 3D, this is caused by the faster breakdown of the tip vortices, while at 5D it indicates a faster transition from near to far wake. On the other hand, in the case of a turbulent inflow, the difference between fixed and pitching cases is once again far weaker, with distributed spots of higher recovery for the pitching cases.

While platform motion does not appear to have a significant impact in terms of mean velocity 3D and 5D downstream the rotor plane, velocity oscillations in the wake appear to be significantly affected by motion. From the perusal of PSD contours (Fig. 13), a significant response at the motion frequency in the LES_t - pitching case is apparent. In addition, the response is not symmetric if the PSDs for positive and negative y/R values are compared. This trend can also be seen in experimental data. While a definitive explanation has not been determined, it is reasonable to assume that the asymmetry stems from the rotor pitching motion, which introduces variations in angle of attack depending on the blade azimuth and rotor position during the pitching cycle due to the clockwise rotation of the blades. These two factors combined result in different angles of attack variations on the left and right sides of the rotor, which may lead to the observed asymmetry. It must be noted that differences between the two sides of the wake are not limited to the velocity spectra, as slight differences in mean velocity values can also be noted at 3D between the two sides of the rotor in Fig. 7. Nevertheless, oscillations at the frequency of 1 Hz – corresponding to the motion frequency of the platform – can be noted even in the LES_t fixed case, especially around y/R=1, even if much smaller than the ones in the pitching case. A 1 Hz peak is noted also in inflow velocity spectrum, which may explain why this frequency can be noted in the fixed case. In addition, the superposition of inflow velocity variation at 1 Hz with platform motion at the same frequency may partially explain why 1 Hz rotor pitch oscillations were found to have a larger effect on wake characteristics than other frequencies in the NETTUNO experimental campaign (Fontanella et al., 2024) or in the UNAFLOW project (Fontanella et al., 2021). In fact, the highest velocity oscillations in the wake were noted when the rotor was oscillating in surge at the frequency of 1 Hz in the UNAFLOW campaign, where the wind tunnel was set at the same operating conditions as those used in this study. A possible interpretation is that a sort of resonance is produced by this superposition of wind turbulence and platform motion. No significant oscillations were noted at the rotor frequency of 4 Hz and at the blade passing frequency of 12 Hz, as such the spectrum is shown up to 3 Hz in Fig. 13. The evolution of the velocity spectra in the wake at 5D downstream the rotor is discussed in section 4.4 and shown in Fig. 30.

In conclusion, rotor pitching motion is found to significantly anticipate tip vortex breakdown and near-to-far wake transition when laminar inflow is considered. Despite the pitching motion introducing velocity oscillations in the wake, the benefits in terms of increased mixing and mean wind speed downstream the rotor are clear. When inflow turbulence is considered, however, even with a moderate 1.5% intensity, the effect of rotor pitching motion is found to be greatly diminished. In fact, inflow turbulence, similarly to rotor pitching, moves the near to far wake transition point upstream. In these conditions, the effect of rotor pitching is still noticeable, but is greatly diminished. Accounting for inflow turbulence in future analyses of FOWTs is therefore essential.

### 4.3 Solving FOWT wake under turbulence: LES or URANS?

LES-based results, discussed in section 4.2, were able to provide insights into FOWT wake development in the presence of inflow turbulence, but came at a significant computational expense, i.e. the pitching case with turbulent inflow required a total computation time of 37 days for 19 s of simulated physical time. Reducing computational costs, while maintaining accuracy in reproducing the involved physics, would be an enabler for future analyses, especially for industrial projects. Therefore, the following sections compare LES simulations to equivalent URANS simulations to evaluate to what extent these methods can still be used to resolve the complex wake dynamics of a FOWT in the presence of inflow turbulence.

### 4.3.1 Fixed case

Time-averaged wake velocity profiles along the horizontal (left) and vertical (right) axes are shown in Fig. 14, while contours of the flow field (in terms of velocity variation with respect to undisturbed flow) on the respective planes are shown in Fig. 15 and 16. Figure 14 shows that the URANS approach is able to qualitatively predict results similar to those generated by LES. Also in this case, in fact, improved agreement with respect to experiments can be noted when inflow turbulence is considered both in the horizontal (Fig. 14(a)) and in the vertical direction in Fig. 14 (b), especially in the top shear layer. Smaller differences between the models can be noted instead in the lower shear layer, where significant interaction between rotor and robot wake takes place (as can be observed in more detail from the velocity contours in Fig. 16). Agreement between numerical models and experiments is anyhow considered satisfactory in this region.

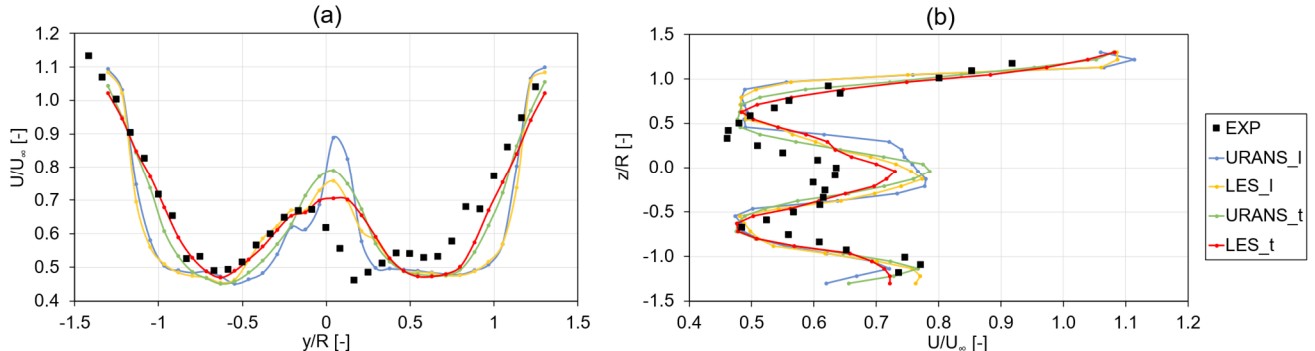

**Figure 14:** Fixed case, wake profiles of velocity at 3D, horizontal (a) and vertical (b) traverse from URANS and LES simulations; comparisons with experiments (EXP).

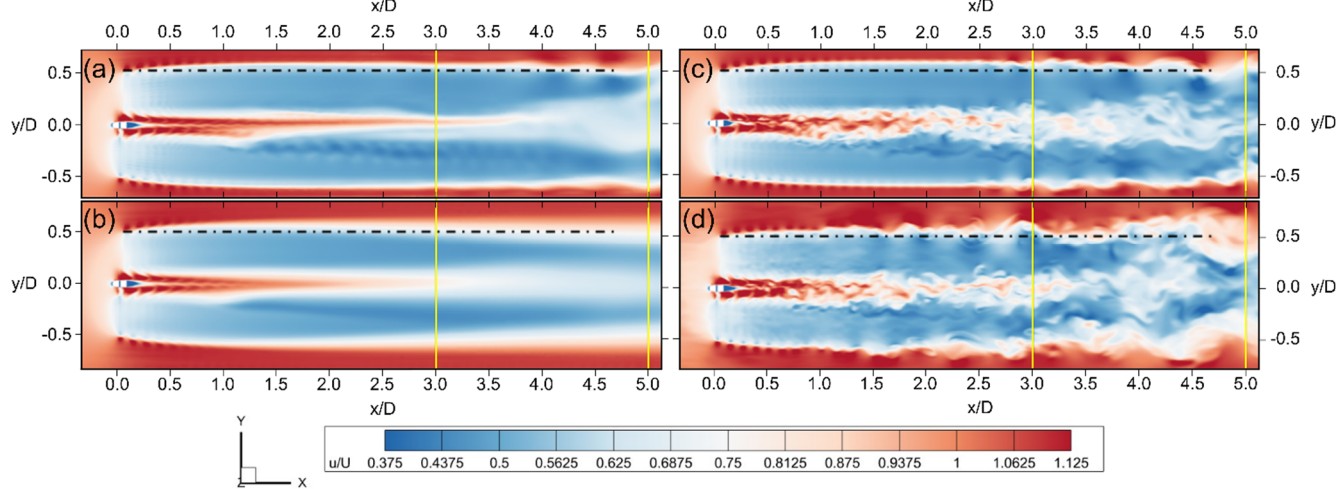

**Figure 15:** Fixed case, velocity contour for URANS with laminar (a) and turbulent (b) inflow, and LES with laminar (c) and turbulent (d) inflow, top view; vertical lines at 3D and 5D. The dashed line represents the upper limit of the turbine's swept area. Distances are reported in diameters from the rotor center.

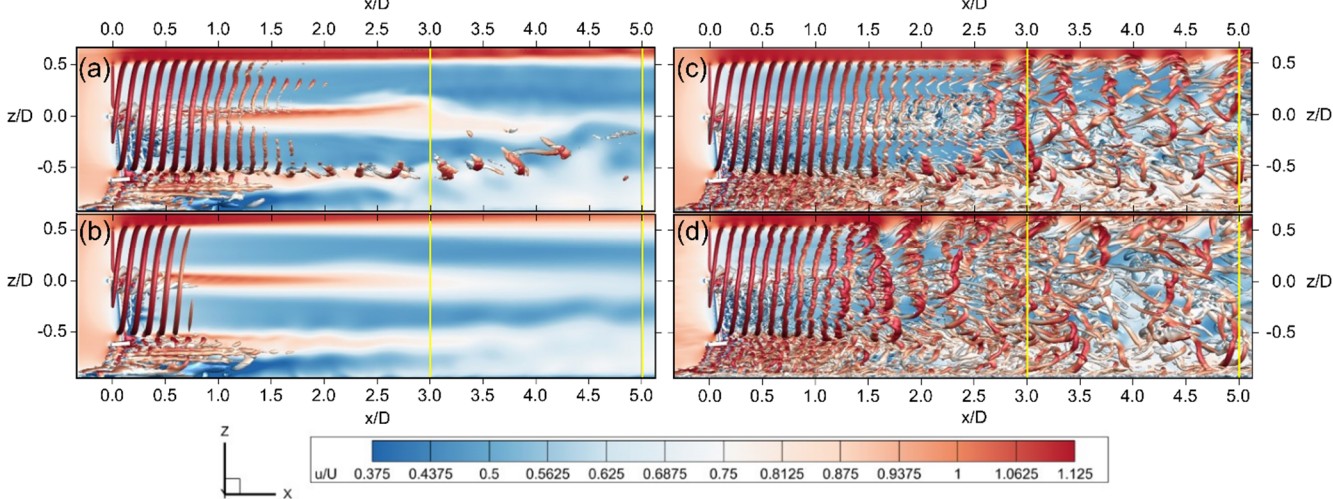

**Figure 16:** Fixed case, velocity contours and iso-surfaces of Q-criterion for URANS with laminar (a) and turbulent (b) inflow, and LES with laminar (c) and turbulent (d) inflow, lateral view; vertical lines at 3D and 5D. Distances are reported in diameters from the rotor center.

With this in mind, it is important to remark that the evaluation of the potential disturbance on the main rotor wake induced by a robot like that used herein (needed to reproduce floating-like motion in wind tunnels) is meaningful when it comes to accurately reproducing the experiments with simulations. In a broader perspective, such an effect is worth consideration in any FOWT system, where the sub-structure generally has considerable dimensions and non-aerodynamic shape, thus able to potentially influence rotor wake.

Shifting attention on contours, good agreement can be noted between URANS and LES in laminar conditions (Fig. 15 (a,c)). Despite URANS simulations appearing able to capture the most relevant flow phenomena close to the rotor, they are apparently not able to preserve the same flow structures downstream the rotor highlighted by LES. Similar considerations also apply to the turbulent case (Fig. 16 (a, b)). However, interestingly the effect of free-stream turbulence is also visible in the URANS simulations, with the tip vortices in Fig. 16 (a) persisting in the flow much longer in the laminar case than in the turbulent one.

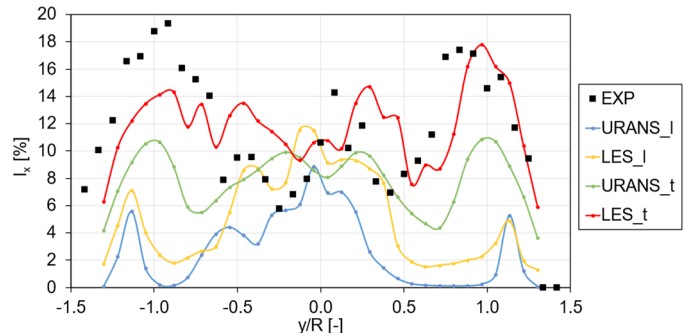

**Figure 17:** Fixed case, streamwise turbulence intensity of the wake at 3D from the rotor, horizontal traverse; comparisons with experiments (EXP).

   Additional evidence of the effects of including free-stream turbulence in numerical models can be gathered from Fig. 17, where

the streamwise turbulence intensity $I_x$ are compared, with the values from LES calculated as per Eq. 2 and the ones from URANS as per Eq. 4:

$$I_{x;URANS} = I_{x;res} + I_{x;mod} = \frac{u'_x}{\overline{U}} + \frac{\sqrt{\frac{2}{3}TKE}}{\overline{U}}. \tag{4}$$

   In the equation, the footer *res* refers to resolved turbulence, calculated by normalizing the streamwise velocity fluctuations $u'_x = std(U - \overline{U})$ by the mean velocity $\overline{U}$, while the *mod* one refers to modelled turbulence, quantified based on RANS-based

turbulent kinetic energy (*TKE*). The turbulence intensity calculated from LES simulations is indeed approximated to a fully resolved one, as the contribution of the sub-grid scale kinetic energy on mean $I_x$ is negligible ($\approx$1%). Upon examination of Fig. 17, the increase in $I_x$ when free-stream turbulence is included in the URANS simulations is apparent. This trend matches the one noted in LES simulations but to a lesser extent, as $I_x$ is underestimated in the outer shear layers in the URANS_t simulation.

Significant differences in terms of mean wake profile (Fig 18 (a)) between the numerical models and the experiments can be finally noted in the central portion of the wake between y/R of -0.5 and 0.5. In this area, URANS results are comparatively similar to LES ones, pointing out that some phenomena taking place in reality are here not captured by the numerical modeling per se.

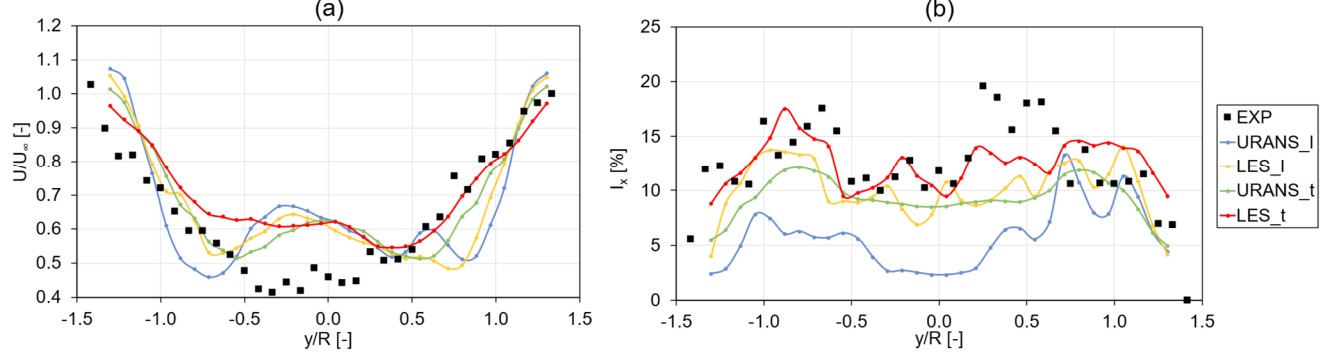

**Figure 18:** Fixed case, mean velocity (a) and turbulence intensity (b) profiles of the wake at 5D from the rotor, horizontal traverse; comparisons between experiments (EXP), reference (REF) results from literature (Firpo et al., 2024) and simulations with turbulent and laminar inflows.

### 4.3.2 Pitching case

Analyses shown in section 4.3.1 have shown that URANS simulations can still provide reliable results whenever mean quantities are concerned, even if turbulence is included. To evaluate if these considerations hold true in the case of FOWT-like motions, simulations with the rotor undergoing pitching motions are compared in this section. Similar to the previous case, Fig. 19 and Fig. 20 first compare the contours of relative velocity, with and without turbulence, in the horizontal and vertical planes, respectively.

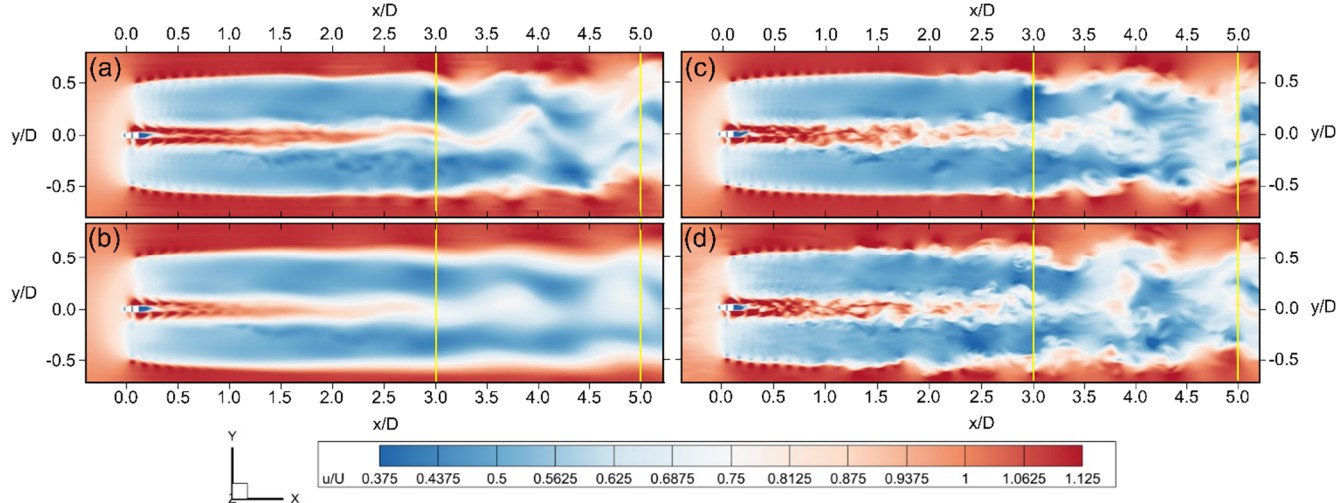

**Figure 19:** Pitching case, velocity magnitude for URANS with laminar (a) and turbulent (b) inflow, and LES with laminar (c) and turbulent (d) inflow at t/Tc = 5, top view at hub height; vertical lines at 3D and 5D. Distances are reported in diameters from the rotor center.

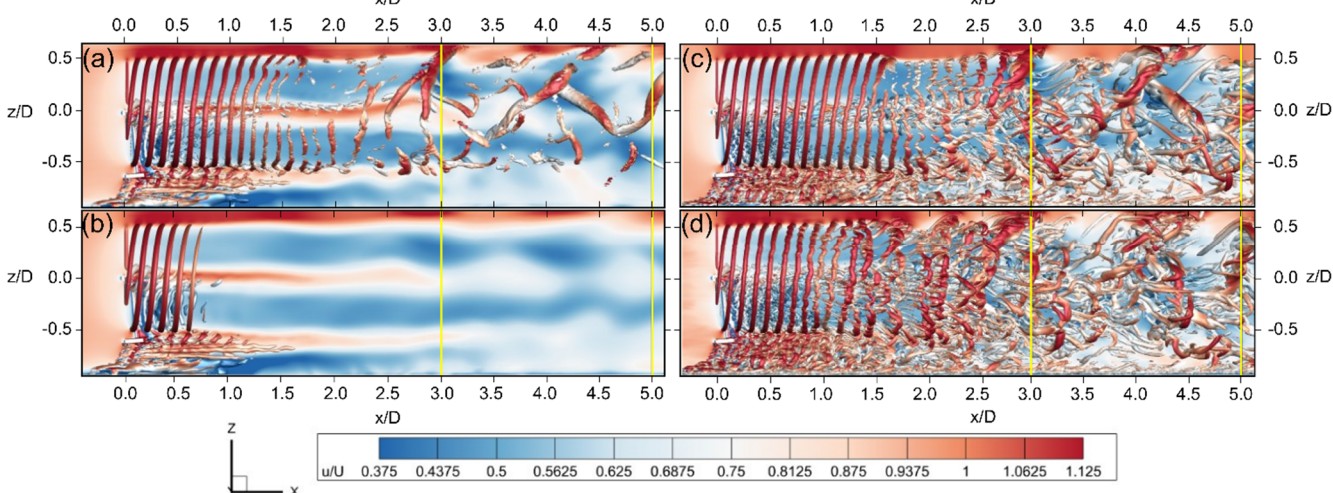

**Figure 20:** Pitching case, velocity magnitude and iso-surfaces of Q-criterion for URANS with laminar (a) and turbulent (b) inflow, and LES with laminar (c) and turbulent (d) inflow, lateral view; vertical lines at 3D and 5D. Distances are reported in diameters from the rotor center.

Upon examination of Fig. 19 (a, b), it is apparent how turbulence has significant influence on the wake even with in the case of URANS approaches, with the URANS_t case showing more significant mixing, less abrupt shear layer, and a quicker decay of the vortical structures. In the URANS_l case, on the other hand, the absence of inflow turbulence brings to a slower vortex breakdown, with large macrostructures that persist throughout the wake; some of these structures are quite similar to those predicted by the LES_l simulations (Fig. 20 (a-c)), with most recognizable ones being the upper vortex at 3D and the two diagonal vortices between 3.5D and 4.5D. The persistence of such macro vortices could also explain the high localized radial gradients in the shear region, with regions with unexpectedly high speed like the one in the lower right flow region of Fig. 19 (a). Looking at turbulent cases, it is possible to notice instead the similarity between URANS and LES velocity contours (Fig. 19 (b-d)) in the main wake patterns, with the exclusion of the external shear regions, where LES can model resolve in a more detailed way the interactions between the turbulent inflow and the pitching turbine wake. These considerations are more evident if the iso-surfaces of Q-criterion (Fig. 20 (b-d)), i.e. the second invariant of the velocity gradient tensor as defined by (DALLMANN, 1983), are compared. As a general remark, velocity contours in the case of a moving turbine look much more similar than those in the fixed case, as further proved by the mean profiles of velocity and turbulence intensity at 3D and 5D shown by Fig. 21 and Fig. 22, respectively. Fair agreement is noted for all numerical models except for some regions (e.g., from -1D to -0.5D), where the inclusion of the free stream turbulence brings results closer to the experiments.

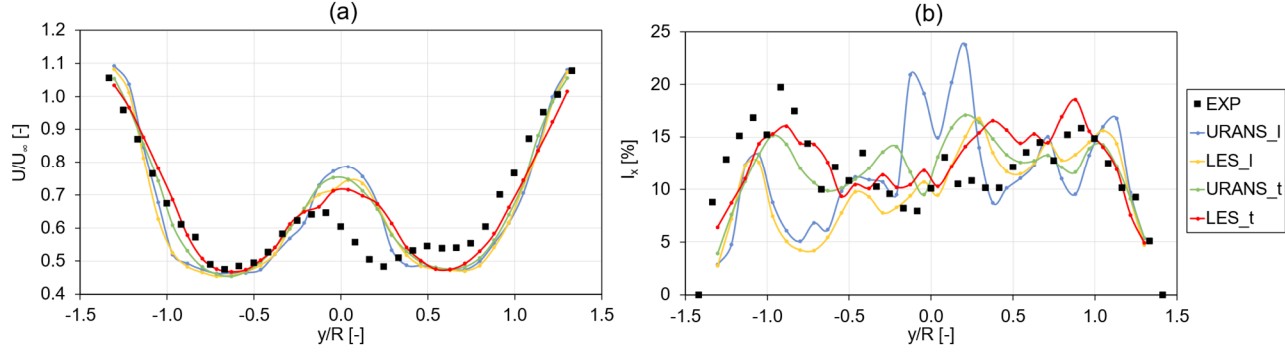

**Figure 21:** Pitching case, mean velocity (a) and turbulence intensity (b) profiles of the wake at 3D from the rotor; comparisons between experiments and simulations with turbulent and laminar inflows.

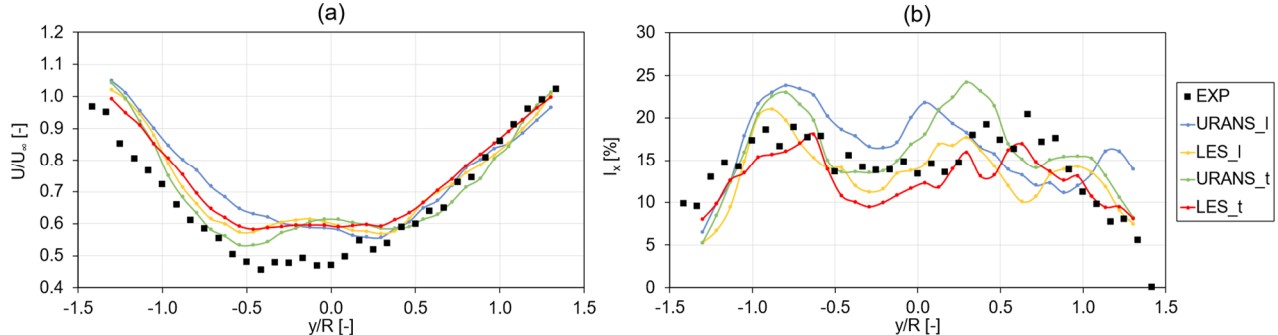

**Figure 22:** Pitching case, mean velocity (a) and turbulence intensity (b) profiles of the wake at 5D from the rotor; comparisons between experiments and simulations with turbulent and laminar inflows.

Interestingly, as the wake develops and moves to 5D distance (Fig. 22 (b)), URANS simulations tend to overestimate the

580 turbulence intensity in most of the wake, while LES simulations are closer to experimental results. At first glance, this outcome could seem counterintuitive if compared with the fixed case, where the streamwise turbulence was always underestimated by URANS simulations. The turbulence induced by the platform motion, superimposed to the modelled inflow turbulence of the URANS_t case, is accurate in terms of turbulence intensity in the near wake, but causes overproduction in the far wake. Similar considerations apply for the URANS_l case, except for some localized overshoots and undershoots noticed at 3D. In any case,

in both LES and URANS approaches relatively little difference is noted in the wake in terms of turbulence intensity between laminar and turbulent inflow conditions. In terms of mean velocity, the inclusion of free-stream turbulence improves agreement of URANS simulations with experiments at 3D (Fig. 21 (a)), while some inconsistencies in the left side of the wake for LES_t and the URANS_l simulation can be noted at 5D (Fig 22 (a)). In particular, the higher wake recovery in the URANS_l simulation with respect to the URANS_t case is counterintuitive, as turbulence should increase recovery rather than diminish

it. For a more in-depth analysis, the standard deviations of the axial velocity are shown in Fig. 23, splitting velocity variations from the modelled turbulence. The extrapolation of the standard deviations shows a high variability of the velocity in the -1.0

< y/R < -0.5 region. Notably, the URANS_l case is much higher than the URANS_t case and more in line with the LES results. This peculiar behavior is likely linked to the steeper shear layer of the URANS_l case with respect to the URANS_t one, as shown in Fig. 19 (a). In fact, as the wake starts meandering and pulsing as a consequence of the rotor pitching motion, sampling points at the wake's edge start to move in and out of the wake, causing larger velocity fluctuations than in the case with inflow turbulence, where the wake shear layer is wider.

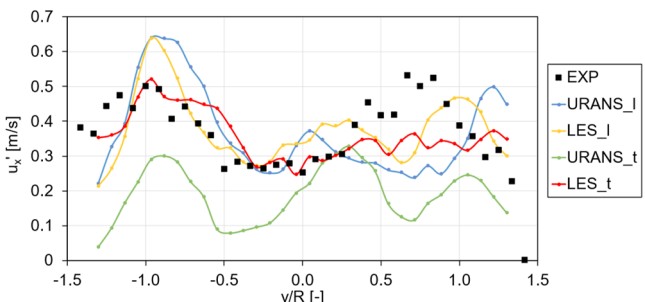

**Figure 23:** Pitching case, standard deviations of the wake velocity at 5D from the rotor; comparisons between experiments and simulations with turbulent and laminar inflows.

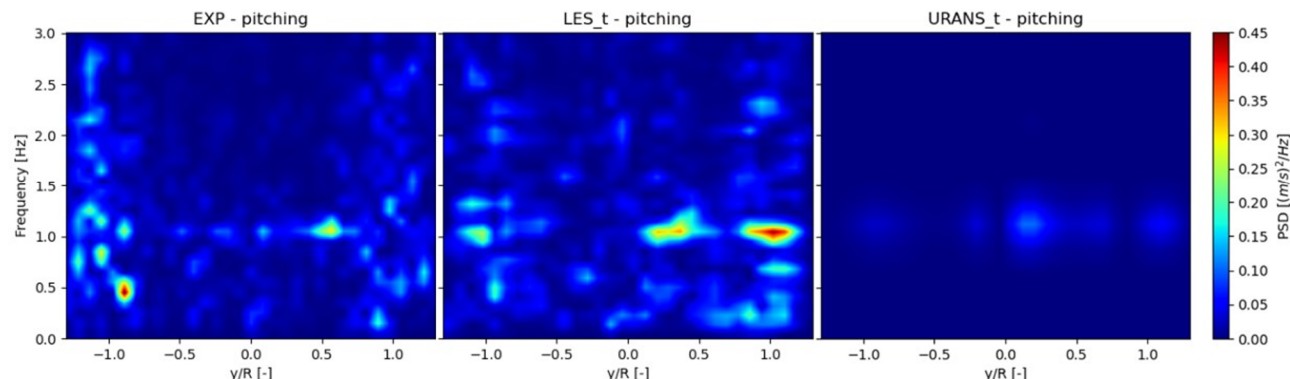

**Figure 24:** Pitching case, contours of PSD calculated from the velocity field sampled at the 3D horizontal traverse; results from experiments (EXP - pitching) and LES (LES_t – pitching) and URANS (URANS_t – pitching) simulations with turbulent inflow; x axis - position on the horizontal traverse, y axis – frequency of each PSD value (colormap).

If the same velocity time series are analyzed in terms of PSD (Fig. 24), other significative differences are outlined between LES and URANS simulations. As already discussed in section 4.2, LES is able to capture the frequency response of the wake well; on the other hand, URANS_t simulations show very low amplitudes, even if at the correct frequencies and positions. This evidence, coupled with the inaccurate wake development highlighted in Figs. 21, 22, and 23, indicates that including the effect of inflow turbulence in URANS simulations by setting appropriate boundary conditions for the turbulence transport equations may not be sufficient for studying FOWT wakes. In fact, this approach eliminates all velocity fluctuations from the inflow, even those that an URANS approach is conventionally deemed capable of resolving. Indeed, as shown in detail in

Appendix A, the inflow turbulence spectrum includes large scale fluctuations, comparable to, or even larger, than the flow features that the URANS approach is considered capable of resolving such as tip vortices. These large-scale fluctuations may interact with the FOWT wake and with the structures created by the rotor motion in a significant matter and neglecting these interactions may be the cause of the incorrect trends highlighted throughout this section.

## 4.4 Hybrid URANS approach

To overcome the discussed limitations of URANS simulations, a novel approach is proposed here in an attempt to combine the strengths of the LES approach with the lower computational cost of the URANS simulations. Large scale velocity fluctuations are inserted into the domain – similarly to LES simulations (section 3.3 and Appendix A) – while maintaining URANS grid requirements and solution. The mesh required for this case features a total cell count of $30*10^6$ elements, roughly one fourth of the cell count required for the resolution of turbulence in LES simulations. Only the pitching case is investigated, in order to evaluate the ability of this method to model the wake structures of a FOWT.

Good agreement is confirmed with the adoption of this *hybrid* method in terms of mean velocity at 3D (Fig. 25 (a)). The two URANS approaches are in good agreement regarding turbulence intensity (Fig. 25 (b)), although $I_x$ is slightly higher in the hybrid approach, possibly due to some overproduction or to some inaccurate estimations in the superposition of modelled and resolved turbulence in the inflow conditions.

Interestingly, at 5D (Fig. 26 (a)), the wake profile given by the hybrid approach is more similar to the LES one. Regarding streamwise turbulence intensity (Fig. 26 (b)), $I_x$ is higher even than the classical URANS approach, although the trend on the y/R axis is more in line with experiments.

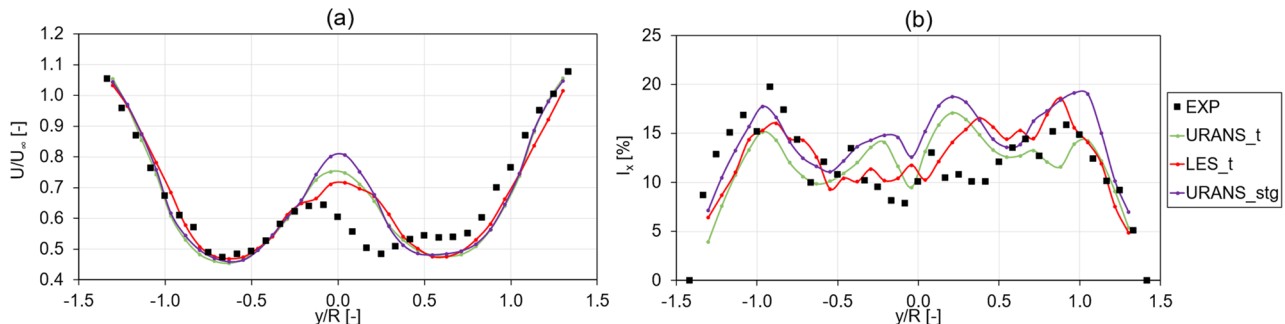

**Figure 25:** Pitching case, wake profiles of streamwise normalized velocity (a) and turbulence intensity (b) at 3D: comparison between experiments (EXP), results from LES simulation with turbulent inflow (LES_t) and from URANS simulation with (URANS_stg) and without (URANS_t) the same turbulence injection adopted for the LES.

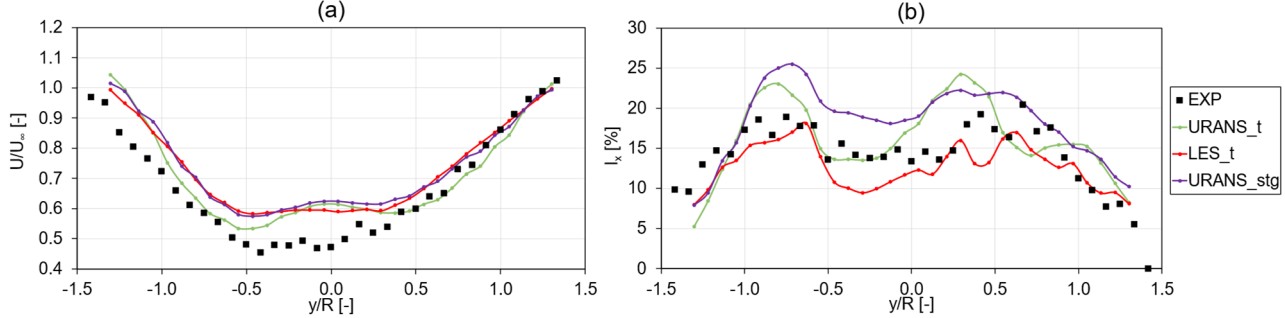

**Figure 26:** Pitching case, wake profiles of streamwise normalized velocity (a) and turbulence intensity (b) at 5D: comparison between experiments (EXP), results from LES simulation with turbulent inflow (LES_t) and from URANS simulation with (URANS_stg) and without (URANS_t) the same turbulence injection adopted for the LES.

Normalized velocity contours and q-criterion iso-surfaces for the pitching case are shown in Figs. 27 and 28. Inserting free-stream velocity fluctuations in the URANS simulations (Fig. 27 (b)) clearly improves agreement with the LES approach (Fig. 27 (c)), as a clear resemblance of the main turbulent structures in the wake between 3D and 5D can be noted between the URANS_stg and LES simulations. The smaller turbulence scales are, however, still absent in the URANS_stg simulation. This

can be seen clearly in Fig. 28, where the URANS_stg simulation, despite once again more closely resembling LES (Fig. 28 (c)), while adequately resolving the tip vortices in the near wake, appears to be unable to solve the smaller structures downstream x/D =1.

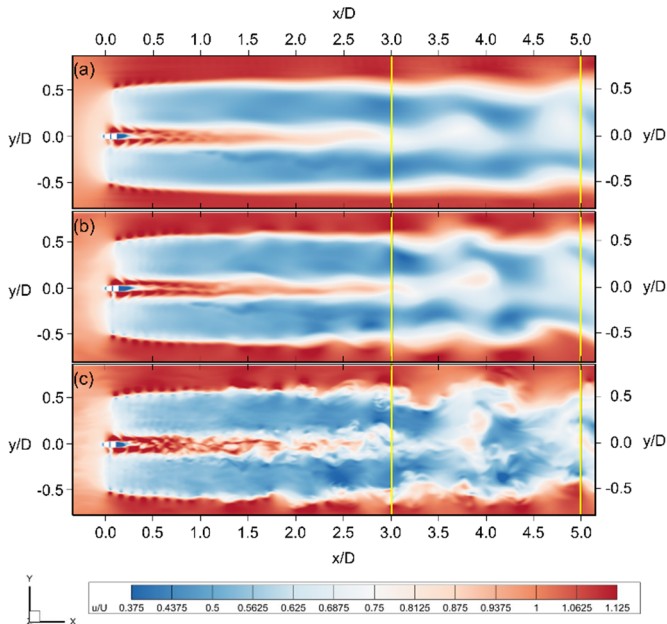

**Figure 27:** Normalized velocity magnitude for the pitching case. (a) URANS (b) URANS-STG (c) LES. Top view at hub height.

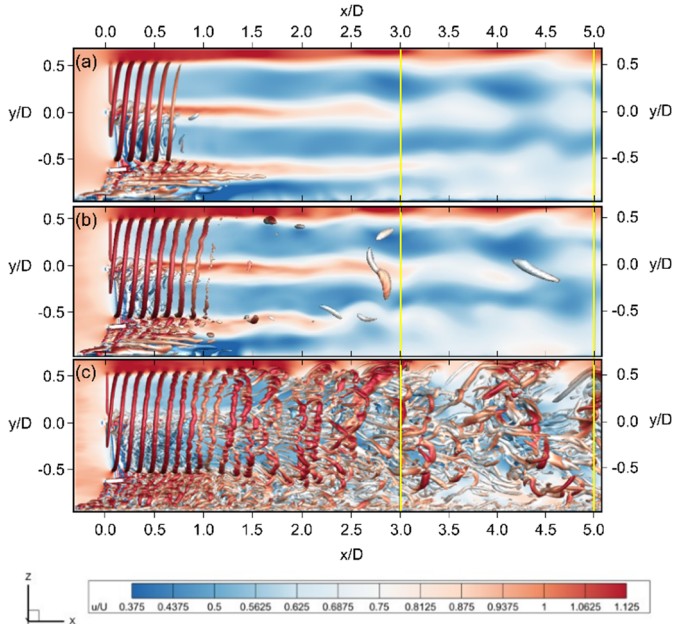

**Figure 28:** Q-criterion iso-surface and velocity magnitude for the pitching case. (a) URANS (b) URANS-STG (c) LES. Lateral view at turbine midplane.

The LES and URANS_stg approaches are compared in Fig. 29 in terms of PSD of the velocity signal. The similarity with the corresponding LES case (central plot) is noticeable, especially in the main spectral components at 1 Hz. Moreover, spectral components in the 0-3 Hz range also appear, and are qualitatively similar to the much more computationally expensive LES simulations. Moving to 5D (Fig. 30), the main frequency response from wake structures is still captured in both space and frequency, but a general overestimation in terms of amplitude is noticed for the hybrid URANS. This result looks in agreement

with the higher streamwise turbulence intensity noted from Fig. 26 (b). To better understand the implications of the differences in amplitude arisen from Fig. 29 and Fig. 30, particularly at the motion frequency of 1 Hz, the mean values and amplitudes of Wake Deficit have been calculated from the horizontal traverses at 3D and 5D according to Equation (5) from (Bergua et al., 2023):

$$WD = \frac{\sum_{i=1}^{N}(U-U_0)|r_i|}{\sum_{i=1}^{N}|r_i|} \tag{5}$$

The index $i: 1: N$ in Equation (5) refers to the measurement points in the horizontal traverses at 3D and 5D.

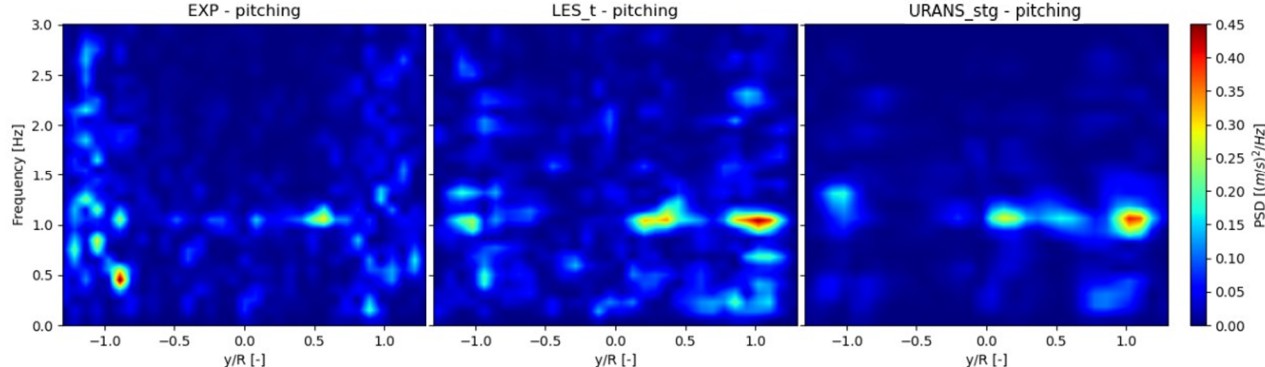

**Figure 29:** Pitching case: contours of PSD calculated from the velocity field sampled at the 3D horizontal traverse. Results from experiments (EXP - pitching) and LES with turbulent inflow (LES_t - pitching) compared to the URANS case with imposed velocity fluctuations (URANS_stg – pitching).

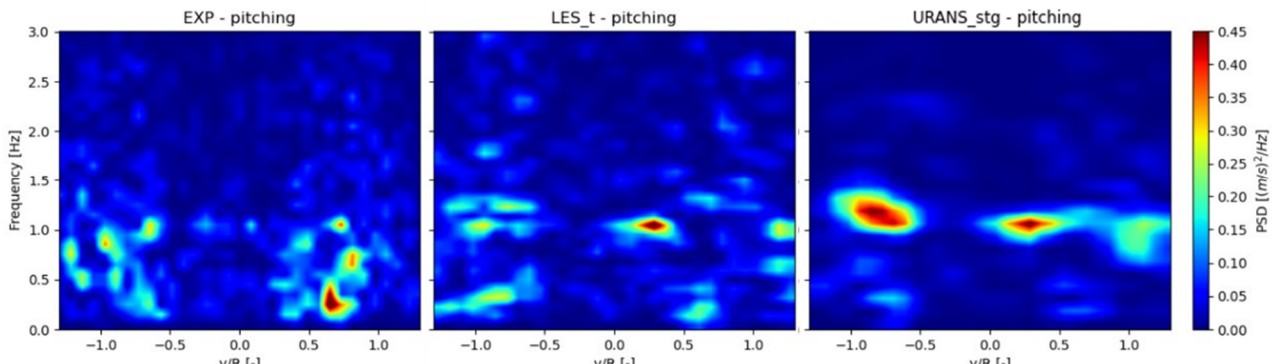

**Figure 30:** Pitching case: contours of PSD calculated from the velocity field sampled at the 5D horizontal traverse. Results from experiments (EXP - pitching) and LES with turbulent inflow (LES_t - pitching) compared to the URANS case with imposed velocity fluctuations (URANS_stg – pitching).

In terms of mean quantities (Fig. 31 (a)), all approaches seem to outline the same trend, despite some differences in the slope of the curve. Similarly to what is shown in Fig. 11, wake deficit decreases from 3D to 5D, and this trend is consistent with the experiments, although overestimated in the numerical models. On the other hand, from the trend of Wake Deficit amplitude $\Delta WD$, calculated as the variation in wake deficit during the phase-averaged cycle (Fig. 31 (b)), some meaningful differences can be noticed. In fact, while experiments show a slight decrease in wake deficit fluctuations (a trend which is well predicted by the LES_t simulation) all other numerical results predict an increase in the oscillation of wake deficit. As also argued by Fontanella (Fontanella et al., 2024), the large coherent structures in the wake are breaking down into smaller eddies as they move downstream, leading to lower overall oscillations in wake deficit at the motion frequency. That being said, two meaningful conclusions can be drawn from Fig. 31 (b). Firstly, the amplitude of Wake Deficit $\Delta WD$ is influenced by free-stream turbulence, as this value tends to decrease moving downstream in the LES_t results, while it increases in the LES_l

result. Indeed, as discussed in section 4.1, turbulence influences the large coherent structures at the motion frequency in the wake, which are more persistent in the laminar case, ultimately leading to the higher oscillations shown in Fig. 31 (b). Secondly, all URANS approaches, including URANS_stg, are unable to capture the experimental trend in $\Delta WD$. Such inability is again linked to the incorrect prediction of the breakdown of the large eddies in the wake. In fact, as discussed in section 4.2, the large coherent structures in the wake are more persistent in URANS simulations than they are in LES, meaning that some wake structures are still dynamically influencing the global wake recovery throughout the traverse span. This conclusion is of highest importance within the scope of this work, as finally demonstrates to what extent the refined resolution of each aspect of the wake – the vortex breakdown, the wake meandering, the turbulent inflow, and its resolution – is directly related to the final accuracy in the analysis of the wake development.

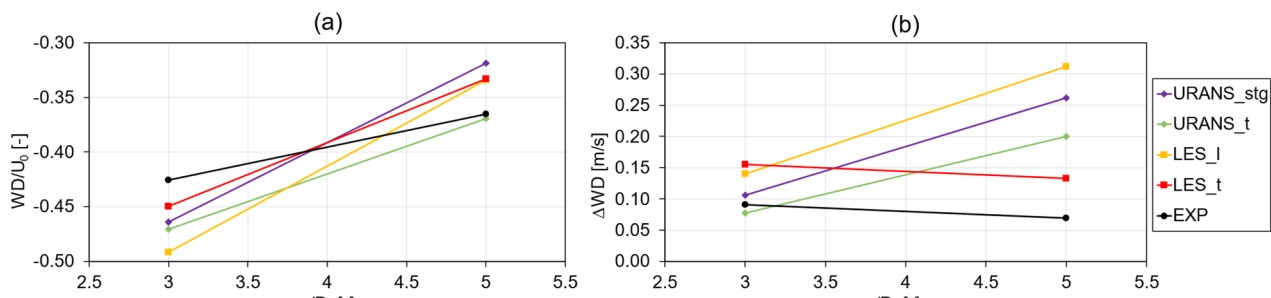

**Figure 31:** Pitching case: mean values (a) and amplitudes (b) of Wake Deficit. Figures show a comparison between results from URANS-STG (URANS_stg), URANS with turbulent inflow (URANS_t), LES with laminar (LES_l) and turbulent (LES_t) inflow, and experiments (EXP). The amplitudes of Wake Deficit are calculated from the phase-averaged cycle, as $\boldsymbol{\Delta WD = WD_{max} - WD_{min}}$ considering the entire sampling time interval adopted for each simulation.

In summary, the proposed hybrid approach is able to provide meaningful information both in terms of mean wake development and resolve the most important frequency components in the wake at a reasonable computational cost if compared to the LES approach. However, the breakdown of the large eddies in the wake caused by rotor motion as they interact with free-stream turbulence is inaccurate and LES simulations appear to be more in line with experimental results.

## 5 Conclusions

In this study a numerical investigation of the influence of turbulence on the characteristics of a FOWT wake is performed. Multiple fidelity CFD approaches are used to solve the wake, ranging from LES to URANS. Results are validated by comparing them to experimental results recently obtained in the Politecnico di Milano wind tunnel. In the numerical campaign, the 1:75 scale experimental rotor is simulated with and without inflow turbulence in fixed conditions and undergoing sinusoidal pitching motion. Numerical tools differ not only in their ability to resolve the wake structures but also in the way free-stream turbulence is accounted for. Indeed, the actual turbulent spectrum measured in the wind tunnel is modelled in the LES simulations, while only mean turbulence intensity and length scale can be included in the URANS approach.

Results have shown that free-stream turbulence affects FOWT wake significantly. Once free-stream turbulence is included in the simulations, the slope of the outer shear layers as recorded in the experimental campaign were correctly predicted by all the numerical approaches. In the central part of the wake, while some differences remain between numerical models and experiments, most likely due to the approximate nacelle geometry in the simulations, free-stream turbulence improved agreement with experiments by decreasing velocity differences in this wake region.

More importantly, the relatively low 1.5% free-stream turbulence intensity for an offshore environment that was included in this campaign, is found to greatly influence the effect of rotor motion on the wake. In fact, while rotor motion is found to accelerate near to far wake transition and promote faster wake recovery in the laminar case, its effect is almost nullified if free-stream turbulence is accounted for at 3D and 5D downstream of the rotor. This result is consistent with experimental observations. Despite this, the signature of rotor motion remains visible in the wake at the reduced frequency of 0.595 that was tested herein.

This result reinforces other recent experimental observations and indicates that free-stream turbulence must be included in numerical simulations focused on FOWT wake development. Although not directly addressed in this study, this conclusion can be also applied – to a certain extent – to future studies on dynamic induction control of FOWTs, which can greatly benefit from inflow turbulence modelling.

In addition, the findings presented in this study suggest that higher turbulence intensities may influence floating wind turbine wakes to an even greater extent. More research is required in this regard to fully understand the effect on turbulence on the wake structures generated by floating wind turbine motion.

In terms of agreement with experimental results, LES simulations appeared the best among the tested approaches, if at a very high computational cost. LES with the addition of free-stream turbulence was able to predict mean wake velocity and streamwise turbulence intensity in the wake at 3D and 5D and is found to agree well with experiments frequency wise, despite some differences emerging at 5D. These differences are imputed to the aforementioned discrepancies in the center part of the wake, which develop downstream and could lead to different wake characteristics at 5D. Despite this, LES is able to predict the correct wake deficit evolution both in terms of mean and oscillation at 3D and 5D, as long as the inflow turbulent spectrum is considered. In fact, it is worth remarking that noticeable differences may arise in this regard if a laminar inflow is assumed. On the other hand, URANS can be used to gain useful information at a fraction of the computational cost. In fact, despite its limitations, this approach is able to clearly highlight the effect of turbulence intensity on the mean wake profile 3D and 5D downstream the rotor. Where the method falls short of its LES counterpart is in the prediction of the evolution of velocity oscillations in the wake, especially in presence of free-stream turbulence. In fact, in terms of frequency content, the only oscillations that are preserved are those at the rotor motion frequency, and at a lower intensity than in the experiments. The inclusion of the turbulence spectrum in the URANS simulations through a LES-inspired boundary condition represents a middle ground between the two approaches and brings URANS results more in line with LES, allowing it to resolve the most important frequencies in the wake. However, the method still falls short of LES if velocity oscillations at 5D are concerned, and it appears unable to correctly predict the evolution and breakdown of the large eddies in the wake as LES does.

In conclusion, this study has confirmed the strong influence of inflow turbulence on FOWT wakes, and proved that numerical tools, if appropriately tuned, can model these effects with varying degrees of accuracy.

## Data availability

Experimental data are openly available at: https://zenodo.org/records/13994980. All data presented in the study are openly available upon request to the contact author.

## Author contributions

LP set-up and performed all simulations, post-processed all results, prepared all figures and data, contributed to data interpretation and first draft. FP provided baseline simulation set-up, conceptualized the study, contributed to data analysis & 755 interpretation and first draft. GC provided software support, contributed to turbulence insertion function implementation and contributed to data analysis. MB and AB supervised the project, contributed to conceptualization, data interpretation and manuscript draft. All authors reviewed the manuscript and contributed to the final draft.

## Competing interests

At least one of the (co-)authors is a member of the editorial board of *Wind Energy Science*. The peer-review process was 760 guided by an independent editor, and the authors also have no other competing interests to declare.

## Acknowledgments

The authors would like to acknowledge Convergent Science for providing the research license of CONVERGE in the framework of Dr. Eng. Leonardo Pagamonci's PhD. Thanks are also due to Dr. Eng. Shengbai Xie from Convergent Science Inc. for providing the original ALM user defined function and for the consulting during data analysis, to Dr. Eng. Pier 765 Francesco Melani for the original formulation of the Line Average sampling method and of the Dağ-Sørensen model that have been adapted and implemented inside CONVERGE CFD framework by Leonardo Pagamonci, and to Dr. Eng. Francesco Balduzzi for the IT support in running the simulations and for the consulting during data analysis. To conclude, many thanks also to Prof. Giovanni Ferrara for tutoring Leonardo Pagamonci's PhD and his support throughout the research project.

## Financial support

We acknowledge financial support under the National Recovery and Resilience Plan (NRRP), Mission 4, Component 2, Investment 1.1, Call for tender No. 104 published on 2.2.2022 by the Italian Ministry of University and Research (MUR),

funded by the European Union – NextGenerationEU– Project Title "Understanding turbine-wake interaction in floating wind farms through experiments and multi-fidelity simulations (NETTUNO)" - Grant Assignment Decree No. 961 adopted on 30/06/2023 by the Italian Ministry of Ministry of University and Research (MUR).

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

## APPENDIX A – Wind tunnel velocity spectrum


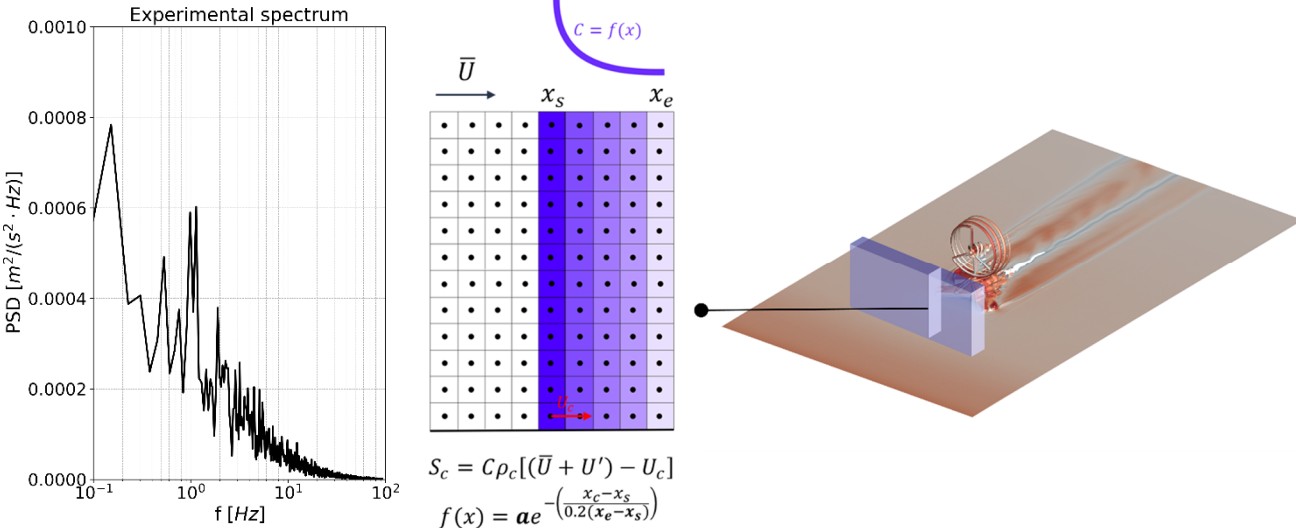

**Figure A1: Experimental spectrum adopted for the turbulence injection inside the LES simulations, and graphical representation of the turbulence insertion box. A slice of the insertion box is represented to show the kernel smearing function C(x).**

The wind tunnel velocity spectrum shown in Fig. A1 in terms of Power Spectral Density (PSD) was sampled before the experimental campaign with the empty tunnel and then repeated aside the turbine installed but with neither rotation nor platform motion. The sampling frequency was 2 kHz. The spectrum has been adopted for the URANS-STG and the LES turbulent simulations presented and discussed in Chapter 4. The wind tunnel spectrum includes a peak at 1 Hz, which corresponds to the motion frequency investigated throughout the paper, although the two phenomena are not directly

connected. The main frequency components extend to a frequency of 10 Hz, where the amplitude of PSD is decreased by a factor of 1/10 with respect to the maximum value.

    A graphical representation of the turbulence insertion zone and the adopted insertion kernel and strategy is shown in Fig. A1. The insertion function Sc can be fine-tuned by adjusting the value of $a$. For the URANS-STG and the LES turbulent simulations presented in this study, the value of 470 is adopted, after preliminary test runs. The turbulence intensity that is obtained at the

rotor plane using this approach is shown in Fig. A2. The average turbulence intensity on the rotor disk is approximately 1.35% in the LES case and 1.2% in the URANS-STG case. The total turbulence intensity, which also accounts for the contribution of the sub-grid scales in the LES simulations and the modelled turbulence in the URANS simulations is approximately 1.5%.

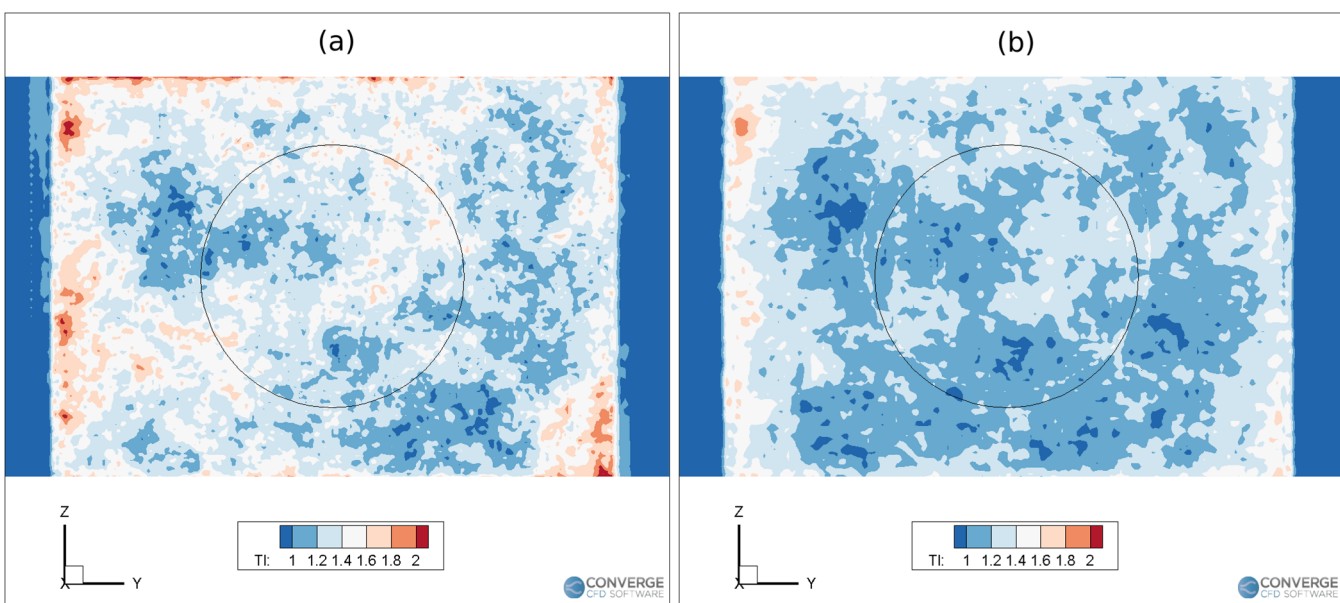

**Figure A2: Turbulence intensity ($TI = stg(U - \overline{U})/\overline{U}$) at the rotor plane in preliminary empty-box simulations without the rotor. (a) LES simulations, (b) URANS-STG simulations. The average turbulence intensity on the rotor disk is 1.35% for the LES simulation and 1.2% for the URANS-STG approach.**


 **APPENDIX B – Mesh resolution in URANS and LES simulations**

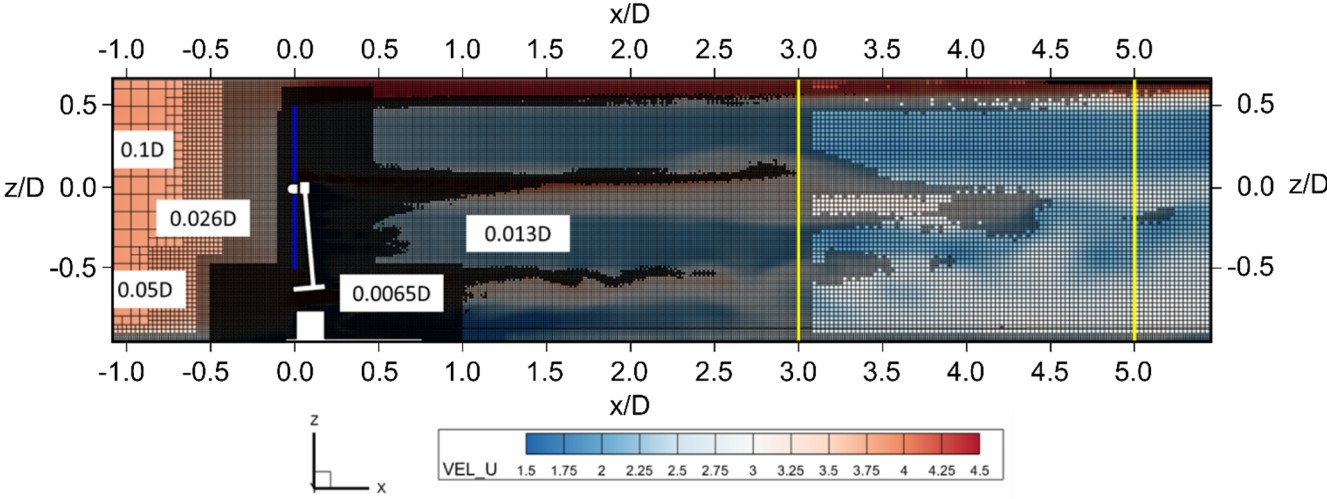

**Figure B1: Mesh setup adopted for URANS simulations, lateral view; yellow vertical lines at 3D and 5D from the rotor and mesh sizes indicated in the labels; red lines – indicative rotor position.**

The domain adopted in this work is discretized using a cartesian grid with progressive levels of refinement (embeddings) close to the rotor. For the URANS simulations (Fig. B1), the cartesian grid with a base size of 0.25 m (≈0.1D) was progressively refined up to the robot, rotor, wake regions, tower, nose and nacelle. For the rotor and the rotor wake, the embeddings have a cylindrical shape, while for the robot a box shape was chosen to completely envelop the entire geometry and accurately solve the wake. The Adaptive Mesh Refinement (AMR) feature available in the flow solver was used to precisely refine the wake

mesh according to the velocity gradient measured in each cell.

    LES simulations required a finer grid than their URANS counterparts, as shown in Fig. B2. Starting from a base grid mesh of 0.32 m (≈0.13D), cells were progressively refined up to 0.01 m (≈0.0042D) of size, around the rotor, tower, robot and root and tip region. For the rotor, which is centered at the origin of the cartesian reference frame, a cylindrical refinement region extending from -0.5 to 1.4 m (≈0.8 D) and with radius 1.5 m (≈0.625 D) was used. The mesh size in this region corresponds

to a $R/\Delta$ ratio equal to 119, which is consistent with the values adopted by other authors for this kind of simulations. In fact, Nilsson et al (2015) (Nilsson et al., 2015) adopted a maximum resolution of $R/122$ at the tip region when simulation the MEXICO rotor, while in the context of the NREL Phase VI, a uniform resolution of $R/100$ was adopted (Churchfield et al., 2017). In a more recent work (Blaylock et al., 2021), the ratio $R/\Delta$ was also higher than 100.

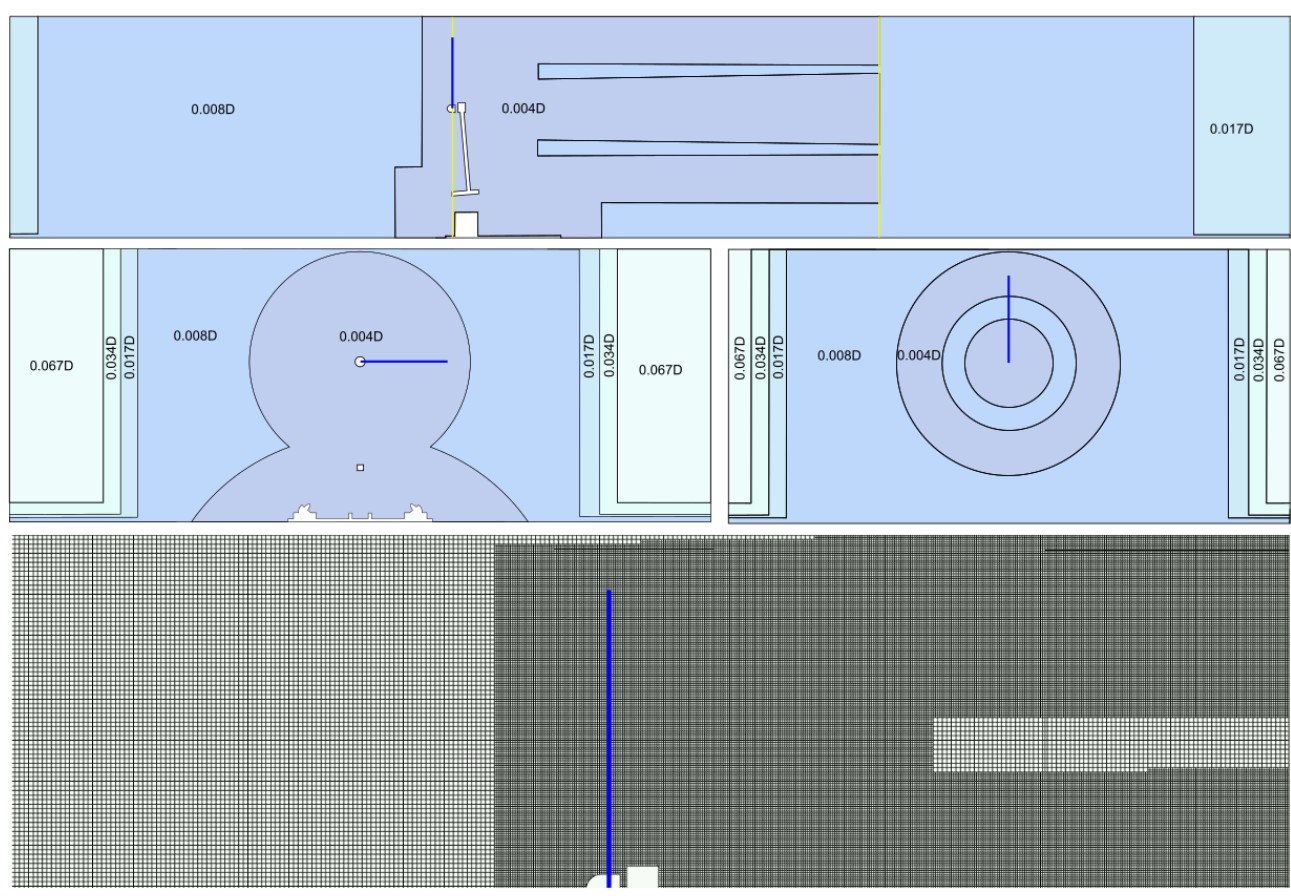


**Figure B2: Top-down: lateral view of the mesh setup adopted for the LES simulations, mesh zones divided by color; front view of the sections (indicated by yellow lines in the lateral view); particular of the mesh adopted for rotor and near wake (blade radius in blue).**

The quality of the LES calculation was one of the key goals of this study. Several quality checks were made both in the sensitivity analyses carried out in preparation of the calculation and during the run. In particular, the resolution in the wake region, which does represent the key element of the study, was monitored every turbine revolution analyzing the contours of the Length Scale Resolution (LSR) parameter. This parameter, introduced by (Piscaglia et al., 2013) for this type of mesh topologies and solver, aims to quantify if the local filter size is small enough to solve the turbulent scales up to the viscosity

range. Values up to 1 do ensure that all the turbulent scales up to the viscosity range are resolved; as per the recommendation of (Piscaglia et al., 2013), LSR values 3-5 can be considered as the limit within which the LES resolution can still be considered as acceptable. Upon examination of Figure B3, which reports a contour of LSR at the beginning of the last revolution simulated herein, in our LES simulation LSR values are consistently lower than one almost everywhere in the domain, including the critical areas of the near wake (within 3D from the rotor) and in the tip-vortex region. This corroborates the selection of

conservative, while computationally-expensive, settings for this study.

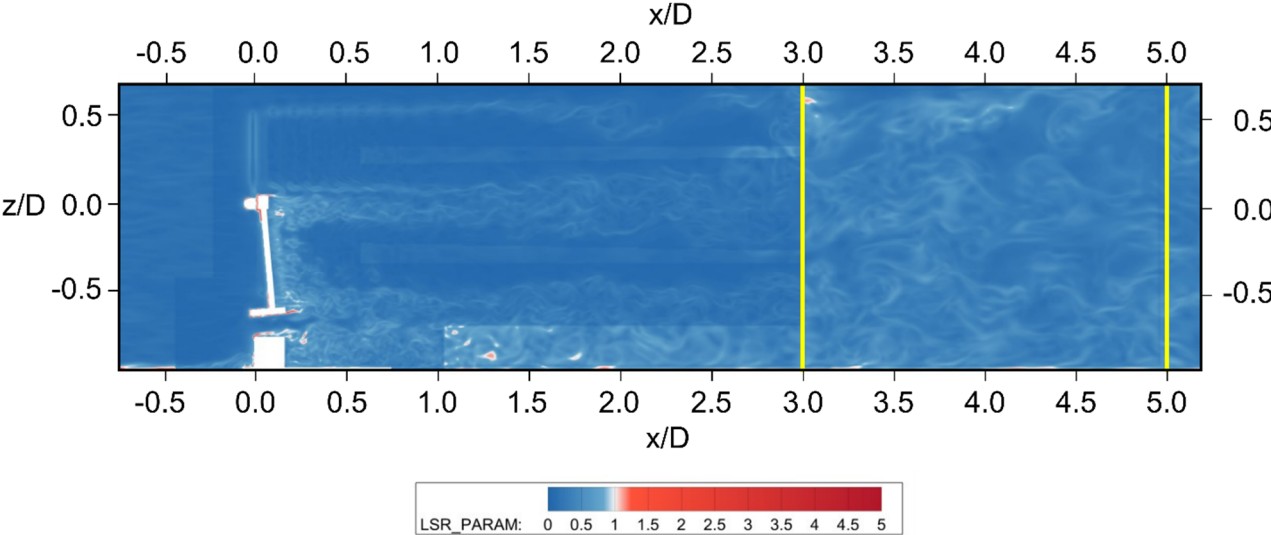

**Figure B3: Contour of LSR parameter at the beginning of the last turbine revolution simulated.**

Finally, to ensure that the injected turbulence is preserved through the computational domain, a box with an internal mesh size
of 0.02 m (0.03125 m for the URANS-STG case) starts from the window 2 diameters upstream the rotor and extends over 5
diameters of distance downstream. In the laminar cases, this region starts approximately 1.3 diameters in front of the rotor,
with the same downstream end. For the resolution of the tip vortexes, a toroidal refinement region, with internal and external
radius of approximately 0.9 and 1.6 meters, was defined and maintained from 1.4 m up to approximately 3 diameters, similarly
to the hub region, where a conical refinement region is defined starting from 1.4 m of distance from the rotor was chosen. The
mesh refinement around the robot, defined here with a cylindrical shape to best follow the geometry and to optimize the cell
count, is again justified by the interest in investigating the possible interaction of the wake detaching from these surfaces with
the rotor wake.
