# Peer review of "How does turbulence affect wake development in floating wind turbines? Some insights from comparative LES simulations and wind tunnel experiments"

_Wind Energy Science, 2024_

## Author Comment (AC2)

**General response to the Reviewers**

Dear Reviewers,

We would like to sincerely thank you for your interesting observations that have made improvements in the paper possible. Based on your comments, we tried our best to improve the paper by clarifying some sections and adding new data and analyses. Modifications have been highlighted in blue-colored text both in the revised version of the paper and in the point-to-point response provided in this document.

We really hope that this revised version can be now worthy of publication in Wind Energy Science.

000 000 000

**Reviewer #2**

Review of "How does turbulence affect wake development in floating wind turbines? A critical assessment". In this manuscript, the authors compare experimental results of wind turbine with platform motion to LES and URANS simulations. The numerical simulations are performed with and without inflow turbulence and the properties of the wake are analysed. The manuscript provides relevant results and unique conclusions on the topic of wakes of floating offshore wind turbines. The manuscript is well-written, however, the title does not fully represent the manuscript and there is lack of description of the methodology. Also, questions can be raised on the flow structures observed in the laminar simulations, as mentioned in the main comments below. Other comments are listed as specific comments. If all comments are addressed, I believe this manuscript will be a relevant contribution to the knowledge on wakes of moving turbines. The authors would like to thank the Reviewer for the time he/she spent revising the study. The qualified observations and the proactive criticism prompted us to re-think part of the study and try to improve it to provide a more robust piece of research.

Main comments:

1. The title of the manuscript promises more than the manuscript deliver. The experiments are performed with only 1 turbulence intensity and the numerical simulations are performed with only 2 turbulence intensities. In my opinion, the manuscript does not fully answer the question proposed in the title, it is only a piece of a larger picture that is still being discovered by the academic community. I strongly suggest a modification of the title.

We do see your point and agree to change the title to more closely reflect the content of the study. The title has been changed into: "How does turbulence affect wake development in floating wind turbines? Some insights from comparative LES simulations and wind tunnel experiments".

2. It is not clear that the flow structures observed on the vortices between 2D and 3D are physical mechanisms, in the laminar case in Fig. 10 (a) and (c). These flow structures look similar to flow structures created by numerical effects. Therefore, the mechanism of vortex breakdown in the laminar case may be due to numerical effects. Please improve the spatial discretization in the wake and show that the results of Fig. 10 are not dependent on the grid.

The Reviewer's point is pertinent. In preparation for this study, where the huge calculation cost allowed for one complete run only, we did our best to assess the accuracy of the calculation settings. While the "RANS-style" mesh sensitivity is not adequate in the case of a LES calculation, we indeed tried a few meshes before selecting the present one, especially regarding the level of embedding (i.e., mesh refinement level) in the wake. The selected mesh was the finest tested, even though another one with one level of refinement less gave already consistent results in terms of flow structures seen in the wake and aggregate performance values. For the reasons explained above, we indeed preferred to stay on the safe side regarding the spatial discretization. Moreover, different from other approaches in the past that made use of progressive wake mesh coarsening, we decided to keep the same level of refinement in the wake from the rotor level up to 3D (see Figure B2), exactly to avoid possible spurious effect due to numerical reasons. In support of the selected settings, we have included in the revised version of the study the new Figure B3, where we report a contour of the Length Scale Resolution (LSR) parameter. This parameter, introduced by (Piscaglia et al., 2013) for this type of mesh topologies and solver, aims to quantify if the local filter size is small enough to solve the turbulent scales up to the viscosity range. Values up to 1 do ensure that all the turbulent scales up to the viscosity range are resolved; as per the recommendation of (Piscaglia et al., 2013), LSR values 3-5 can be considered as the limit within which the LES resolution can still be considered as acceptable. Upon examination of Figure B3, which reports a contour of LSR at the beginning of the last revolution simulated herein, in our LES simulation LSR values are consistently lower than one almost everywhere in the domain, including the critical areas of the near wake (within 3D from the rotor) and in the tip-vortex region. This corroborates the selection of conservative, while computationally-expensive, settings for this study.

3. In Appendix A, is the velocity sampled from an experiment with platform motion or without motion? If the velocity is sampled from a case with motion, then the simulations of the "fixed" turbine with turbulence will be contaminated

by the frequency of motion. If this is the case, this is an important limitation of the study and should be clearly discussed in the results.

The Reviewer's comment is surely correct. Fortunately, this is not the case of our study. The velocity was sampled within the tunnel without platform motion and even without turbine rotation. The spectrum was also compared to that measured before the tests without the experimental setup, obtaining total agreement. A discussion has been added to the paper. By double-checking the text, we indeed noted that it could be misleading and are sorry for this. The text below Figure A1 has been rephrased.

Specific comments:

1. Please define all abbreviations when they first appear. Many of the abbreviations are not defined, some examples being: ALM, URANS, BEM, DOF (list not extensive). AMR was only defined in the appendix.

The Reviewer is perfectly right. All acronyms have been defined at their first occurrence in the text.

2. In line 66-65. (Xu et al., 2024) is mentioned as a paper that suggested that the turbulence change some of the phenomena described so far in numerical studies. However, this work did not only suggest this, it studied this effect. Please rewrite it making clear the contribution of the cited article.

Done. We kept the text short for brevity, but indeed the explanation was incomplete.

3. In line 95. Apparently there is a missing reference ("[X]"). Thank you for noting. No reference was needed there.

4. In section 2, please add the Reynolds number and temperature of the experiment. Added at the beginning of section 2.

- In Table 2, please add the tip speed ratio and indicate the rate of pitching frequency to rotor frequency, which affect the growth of instabilities according to Kleine et al, 2022.
  Added.
- 6. The solver is not mentioned in the section 3.1, only in section 3.2. Please include the name of the solver, references and a brief description in section 3.1.

Correct. We have moved the description of the software in 3.1.

7. Please reorganize section 3.1, dividing the section in smaller sections (or subsections), to make clear the parameters of each simulation and numerical method. Please provide a brief description of each numerical method and include references.

Section 3.1 was split into a section describing the CFD domain and boundary conditions, and one describing the grid sensitivity study.

8. In line 122. The term "nose cone" is more specific than "nose". Thank you for pointing this out. Added.

9. The use of the term URANS-Hybrid is very confusing. I was only able to understand it after reading section 4.4, hence it should be better described in the methodology. More importantly, it is not a hybrid between URANS and LES, as it could be understood from context. It is a URANS simulation with a different strategy of imposing inflow turbulence, without using any of fundamentals of LES. I do not believe the name is adequate.

We agree with the Reviewer. Although we intended the term "hybrid" in the sense explained at the beginning of section 4.4, it can be confusing. We have thus renamed this approach "URANS-STG" as the inflow velocity fluctuations are generated using a synthetic turbulence generator. We have better described the approach in the introduction and also in section 3. The approach that is used to account for free-stream turbulence is also described in section 3.1 and briefly repeated in section 4.4.

10. Please define what is the AMR threshold and provide a reference for the AMR.

A more complete definition of the adopted threshold has been reported in the paper, and a reference is provided.

11. Please indicate the distances from the boundaries to the rotor.

The wind tunnel inlet is placed 8.8 rotor diameters upstream of the rotor. The domain extends up to 6 diameters downstream. These boundary placements were defined based on the wind tunnel test section, which extends a corresponding amount upstream and downstream the rotor. The lateral walls of the tunnel are placed 2.8 diameters away from the rotor center, again based on the physical dimensions of the wind tunnel.

12. In line 189. Please indicate the width of the Regularization Kernel clearly in the manuscript.

The width of the regularization kernel varies along the blade span. In particular, the local width of the kernel is chosen based on (Xie et al., 2021). In particular, the kernel size equals the local chord length, with a lower limit based on cell size:  $\beta = \max(c/4, \zeta \Delta)$ . In addition, since the grid size in the LES and URANS simulations is different, kernel size was fine-tuned to the specific approach, with the purpose of maintaining – in all the simulations - the same spatial force distribution throughout the computational domain. Therefore,  $\Delta = 0.01$  m and  $\zeta = 2.4$  in the LES simulations and  $\Delta = 0.015625$  m and  $\zeta = 1.6$  in the URANS simulations. This choice was adopted to ensure that the blade representation remains as consistent as possible in the compared simulation approaches. The reason for this is twofold, firstly the blade model was verified during the OC6 Phase III comparative campaign, with good alignment with respect to experimental results and other numerical models. Secondly maintaining a consistent blade model ensures that the compared wake solution methods can be compared. This discussion has been reflected in the paper in section 3.3.

13. In line 97. There is a mistake in equation 22 of (Dağ and Sørensen, 2020), as pointed out by (Meyer Forsting et al., 2019) and (Kleine et al., 2023). Please indicate clearly how the correction velocity was calculated. References: [1] Meyer Forsting, A. R., Pirrung, G. R., and Ramos-García, N. "A vortex-based tip/smearing correction for the actuator line." Wind Energy Science 4, no. 2 (2019): 369-383.; [2] Kleine, V. G., Hanifi, A., and Henningson, D.S. "Non-iterative vortex-based smearing correction for the actuator line method." Journal of Fluid Mechanics 961 (2023): A29.

The Reviewer's comment is on point, and the first draft of the paper was indeed not sufficiently clear on this aspect: our apologies. When we first implemented the tip correction for our ALM code, we did it for a simple wing, thus the additional induction term due to the wake structure (cited eq. 22 in the original paper) was not needed and only took the basic structure of the Dağ and Sørensen model. This is where the original test on the ALM was partially copy/pasted in the first draft. When introducing the model in Converge, we instead added the term, and we did it with the formulation of Meyer Forsting et al. (your suggested ref. [1]). The authors were aware of the work of Kleine et al., but no attempt to implement it has been made so far. The paper has been rephrased to clarify.

14. Section 3.3 is not clear. Please define mathematically the desired velocity perturbation, the exponential distribution function and the input velocity perturbations. Include more details on the generation zone.

The desired velocity perturbation  $\Delta u = U - U_{x;\infty}$  is the difference between the desired velocity accounting for turbulent inflow fluctuations U and the undisturbed velocity  $U_{x;\infty}$ . Additional details on the generation fzone, exponential distribution function have been added to section 3.4 and to Appendix A. The input velocity perturbations are generated using the synthetic turbulence generator TurbSim with the velocity spectrum as input. Additional details can be found in section 3.4.

15. Please present the the measured properties/statistics that shows that the position of the generation zone of turbulence is adequate for both the LES and URANS ("hybrid") simulations.

Mean velocity and turbulence intensity  $(TI = stg(U - \overline{U})/\overline{U})$  have been sampled from preliminary empty-box simulations (i.e. without the rotor), specifically designed to verify the insertion of turbulence perturbations. These results have been added to Appendix A, and referenced in section 3.4.

16. In line 220 and appendix B. Some distances are indicated in meters, without indicating non-dimensional distances. Non-dimensional distances have been added.

17. In section 3.4 and figure 12. Please indicate the sampling window in number of cycles instead of seconds. Please also indicate the number of rotor revolutions.

Thank you for pointing this out. We have indicated sampling windows in seconds because the discussed windows apply both to simulations with and without pitching motion, where defining the sampling window based on pitching cycles is inconsistent. We have, however, added additional details to better clarify this point in section 3.5 of the revised manuscript.

18. Please define  $\Delta$  in tables 5 and 6.

The tables and the table legends were amended to better clarify this.

19. In line 292. What do you mean by left part of the wake? Positive or negative y? The text was unclear. Corrected.

20. In figure 9, reference to iso-surfaces. The caption has been corrected.

21. In figures 9, 10, 15, 16, 19 and 20. Include the coordinates in every axis. Coordinates were added in terms of distance from the rotor center, in rotor diameters.

22. In equation 3. Probably there is a typo. Is it non-dimensionalization by area? Which is the integrated area?

Thank you for noting the typo, the area has been included in the non-dimensionalization. The integrated area -more specifically, the rotor area - has been specified in the text.

23. Please define the differential wake deficit mentioned in the caption of Fig. 12.

The Reviewer is indeed right, and this aspect was too vague in the first draft. For best clarity, both the definition of the wake deficit adopted in this section and the definition of the differential wake deficit have been included in the text.

24. Please indicate clearly the frequency of movement, the blade passing frequency and the frequency of the rotor. Frequency of movement was added to figure 13. Rotor frequency was added to Table 2. No significant trace of the rotor frequency (4Hz) or the blade passing frequency (12 Hz) was noted in the wake, thus the higher frequencies are omitted from Fig. 13.

**25. Please explain why the results from Fig. 13 are far from symmetric.**

The Reviewer raises an interesting point. PSDs are indeed asymmetric even in the experimental data, which indicates the good ability of the simulations to reproduce the physical behavior of the test case. While a definitive explanation of the phenomenon has not been determined, hence the reason why this discussion was omitted from the first version of the manuscript, it is likely due to the combination of clockwise rotation with rotor pitching motion. In fact, rotor pitching motion introduces variations in angle of attack depending on the blade azimuth and rotor position during the pitching cycle. This ultimately results in different angles of attack variations on the left and right sides of the rotor, which may lead to the observed asymmetry. The discussion of Fig. 13 was improved to reflect this.

26. Show that the PSD converged. Convergence of the mean values (discussed in section 3.4) does not imply convergence of all statistics.

We agree with the Reviewer in the fact that convergence of means does not imply full statistical convergence. Mean values are shown in section 3.4 as these are used to evaluate wake recovery, which is an important point of discussion in floating wind turbine wakes, and one of those we have highlighted throughout the paper. Nevertheless, in preparation of the study, we made several checks on the convergence of our calculation. In particular, the PSD at 5D downstream the rotor is shown at r/R=0.71, where the mean values in Fig. 4 are slowest to converge and at r/R = 1.05 where the spectrum shown a peak in Fig. 13. As shown below, good convergence can be noted. To reflect this point, a comment was added to section 3.4, and the standard deviations were added to the mean values in Fig. 4. Good statistical convergence can be noted for this statistical moment as well.

**27. Do the results from Fig. 13 change as the position move downstream? Can the same peaks in frequency be observed downstream?**

The Reviewer highlights an interesting point. The spectra shown at 3D in figure 13 are shown at 5D in Figure 28. The spectra at 3D are also partially repeated in Fig. 27, which allows for a more convenient visual comparison with Figure 28. We decided to "condense" the results this way to keep the paper more synthetic. As discussed in section 4.4 as we move downstream, the peaks at 1 Hz can still be seen, but low-frequency peaks also start to appear. We attribute the latter to the breakdown of some of the coherent wake structures.

**28. Please show the PSD at 5D.**

PSDs at 5D are shown in Fig. 28. We preferred to include this here to avoid redundancy. A comment has been added to the text to reflect this in section 4.2.

29. In lines 395 to 397. The explanation for the presence of a peak in 1 Hz for the fixed turbine is not clear. Please provide a more detailed explanation.

The peak in the spectrum at one hertz is believed to originate from the peak in the inflow velocity spectrum at this frequency, a shown and discussed more clearly in the revised Appendix A. The discussion of Figure 13 was reworked to address the Reviewers' comments.

30. In section 4.4. Please show figures similar to 19 and 20.

The Reviewer's request is pertinent. Indeed, we decided not to include the pictures in the first draft only for brevity. However, to ensure completeness, snapshots with results from URANS\_t, URANS\_stg and LES\_t have now been included.

31. In Figure 29(b) and page 27. The definition of  $\Delta$ WD is not clear.

The Reviewer is right, and the definition was not completely clear.  $\Delta$ WD is calculated as an average - over each point along the horizontal traverse - of the amplitudes of wake deficit variations. Additional details have been included in the paper for better clarity.

32. Figure A1. Caption is probably incorrect. The Reviewer is right. Probably, there was a copy/paste issue. The caption has been fixed.

33. Appendix A. Where was the velocity sampled? Details have been added as requested.

---

## Author Comment (AC3)

**General response to the Reviewers**

Dear Reviewers,
We would like to sincerely thank you for your interesting observations that have made improvements in the paper possible. Based on your comments, we tried our best to improve the paper by clarifying some sections and adding new data and analyses. Modifications have been highlighted in blue-colored text both in the revised version of the paper and in the point-to-point response provided in this document.
We really hope that this revised version can be now worthy of publication in *Wind Energy Science*.

ooo   ooo   ooo

**Reviewer #1**

Overall, this is an excellent paper, and I would like to sincerely congratulate you on your work. I really enjoyed reading it and have just a few minor comments.
Dear Reviewer, thank you for your appreciation of our study. We did our best to further improve it based on your comments.

- Can you highlight how often/realistic the conditions are offshore that are being studied?
Thank you for the right comment. The amplitude and frequency of the studied pitch motion translates to a full-scale oscillation of 1.3° amplitude and 25s period. Full-scale to model-scale conversions can be done by imposing the conservation of the reduced frequency $f_r = f * \frac{D}{U}$. The oscillation corresponds to a relevant oscillation close to the design natural frequency of many floaters, and is thus considered quite realistic. A comment has been added to section 2.
Regarding the inflow, a turbulence intensity of 1.5% can instead be considered a low value inside the atmospheric boundary layer. Despite the low value, turbulence is found to affect wake characteristics very significantly. We assume this effect to be even more pronounced at higher turbulence intensities. This discussion was reflected with an additional comment in the conclusions.

- Abbreviations are not always explained, for instance NETTUNO (line 14), URANS (line 22), BEM & CFD (line 56), ALM (line 126), HAWT (line 196)
The Reviewer is right. We have gone through the paper and defined all acronyms. However, since they are well-known, we preferred to define them within the text rather than in the abstract for conciseness. LES only is defined also in the abstract as it used gain therein.

- Line 40: The word 'the' missing before 'system'
Corrected.

- Line 68: 'Turbulence in (typo?) not explicitly solved' - can you please explain what this means in an extra sentence?
The Reviewer is right. Beyond the typo, while this concept is probably easily arguable by experts in the topic, a better explanation was due to make the discussion clearer to all readers. We have expanded the sentence.

- Line 94: 'Share' not 'shares'
Corrected.

- Line 139: Please explain 'law of the wall'
Done.

- Line 156: Please explain 'bottom shear layer'
The term refers to the lower part of the wake's edge, where the faster surrounding flow mixes with the slower-moving rotor wake. We have clarified this better by specifying "the bottom shear layer of the wake"

- Line 229: 'LES_t' indexing not explained until the next page
Right. The definition of the different simulations has been concentrated at the beginning of Section 4.

- Table 6: Please explain what the +/- values are (standard deviation?)

The reported values represent the maximum and minimum deviations from the mean value of the phase-averaged variations of rotor thrust and torque. The phase-averaging process refers to averaging multiple pitching cycles in order to obtain the "mean" cycle. An explanation for this was added to section 3.5.

- Equation 2: Please explain all the variables
Explained

- Very small and partially covered coordinate system axis description in Figures: 9,10,12,15,16,19,20,B1
Also in response to the comments by Reviewer #2, coordinates were added in terms of distance from the rotor center, in rotor diameters.

- Line 354: 'But then' grammar mistake
The Reviewer is perfectly right. The sentence has been rephrased.

- Equation 3: Typo in U_0 numerator
Corrected.

- Figure 15: Please explain the dotted line at the top
Added.

- Equation 4: Please explain u'x
u'x is the standard deviation of the velocity fluctuations: $u'_x = std(U - \overline{U})$. The clarification has been added to the text.

- Line 493: An Equation to show the velocity gradient tensor would be nice
The sentence was indeed misleading here. We have rephrased the comment to Figure 19. With this formulation, we think no equation is needed anymore.

- Line 514: The word 'with' is missing before 'respect'
Added.

- Line 547: Please explain why the pitching case is the most effective test case to be considered
This line was somewhat confusing. We have tested the pitching case only as we are most interested in the ability of the proposed method to resolve the wake structures typical of a floating wind turbine. We have changed the text to reflect this.

- Equation 5: Please explain N (I am assuming it is the number of sections the blade is discretized into?). Is U0 the same here as U_0 in Eq. 3?
U0 is the same as U:0, this was a typo and has been corrected. The index i:1:N refers to the measurement points along the horizontal line at hub height at 3D and 5D from the rotor. This has been better clarified in the paper.

- Line 627: Please add units for the frequency (Hz)
The value is reported in terms of reduced frequency, so no unit is needed.

- Appendix A, line 814: Please explain in more detail why there is a peak at 1Hz.
The velocity spectrum reported in Fig. A1 was measured in the empty wind tunnel. The measurement was then repeated with the presence of the turbine with neither rotation nor platform motion. In the latter case the anemometer was offset with respect to the turbine, i.e. not placed directly behind it. The reason a 1Hz peak is visible in the spectrum is not fully understood but is most likely a result of the turbulent structures that naturally tend to form in the wind tunnel. The text was amended to better explain the nature of the spectrum.

---

## Author Response (AR2)

**Response to the Editor**

I regret to write you that one of the reviewers are still not fully satisfied with your revised paper. As I understand it, the main issue is that the reviewer wants you to demonstrate that the instabilities causing vortex breakdown in the laminar case is not due to numerical issues: *'It is not clear that the flow structures observed on the vortices between 2D and 3D are physical mechanisms in the laminar case in Fig. 10 (a) and (c). These flow structures look similar to flow structures created by numerical effects. Therefore, the mechanism of vortex breakdown in the laminar case may be due to numerical effects. Please improve the spatial discretization in the wake and show that the results of Fig. 10 are not dependent on the grid.'*

I suggest that you to carry out the proposed additional computation, if possible, as this is a cornerstone in the analysis. If this is not possible, you need as a minimum to discuss it in the paper. Finally, there are suggestions some minor modifications, that should be easy to implement.

Dear Editor, thank you for your coordination of the review process. We are anyhow grateful to the Reviewer as constructive criticism is essential to improve the scientific quality of the studies. While an additional calculation is not at hand, we have included significant new discussion in the paper to address the doubts raised by the Reviewer. We hope that the new version of the paper can be now worthy of publication in WES.

ooo   ooo   ooo

**Reviewer #2**

Review of "How does turbulence affect wake development in floating wind turbines? Some insights from comparative LES simulations and wind tunnel experiments". The authors satisfactorily addressed most of the previous comments in this version of the manuscript. There has been a clear improvement. My main concern is related to the old main comment #2, which I consider still open. Since this is a critical point, I reinforce the need for more studies to show that the results are not dependent on the numerical method. Additionally, there are a few remaining specific comments that I recommend addressing before publication. The comments that I consider to be still open are listed below:

**Main comments:**

1. (Old comment #2. It is not clear that the flow structures observed on the vortices between 2D and 3D are physical mechanisms in the laminar case in Fig. 10 (a) and (c). These flow structures look similar to flow structures created by numerical effects. Therefore, the mechanism of vortex breakdown in the laminar case may be due to numerical effects. Please improve the spatial discretization in the wake and show that the results of Fig. 10 are not dependent on the grid.)

I appreciate that the authors took steps to select the mesh. However, unfortunately, I do not consider this issue solved. This is a major issue that could raise questions regarding how well the simulations reproduce the physics of the flow. The second half of the authors' response is focused on the Length Scale Resolution, which is a metric related to turbulence and the resolution of small turbulence scales. Maybe I was not clear in my comment. This is an issue of stability and numerical perturbations, not turbulence. When performing a simulation without imposing any perturbation, the numerical method itself introduces perturbations in the form of numerical errors. These numerical perturbations commonly have very specific frequencies and wavenumbers, usually related to spatial and/or temporal resolution. These numerical perturbations may grow in the "laminar" part of the flow, forming flow structures that would not occur in a real flow. The flow structures seen in Fig. 10(a) and (c), between 1.5D and 2.5D, resemble some of these numerical flow structures. My suggestion is that the authors improve the spatial discretization for the case shown in Fig. 10(a), keeping the same regularization kernel width (not the same \zeta). If the flow structures around x/D=2 are modified, then there is an indication that the numerical method and numerical parameters are affecting the results.

We appreciate the additional details included by the Reviewer that allowed a further critical analysis of our work. Unfortunately, we are unable to run a finer mesh as the computational resources that we used (the calculation has been run on a national super-computer for which a competitive grant was obtained) are not currently available to us. The use of such calculation infrastructure was needed because the calculation is very demanding.

We do understand the concern of the Reviewer and respect his/her scientific doubts: we have tried our best to address them in the paper better now that they are clearer to us. The flow structures between 1.5D and 2.5D resemble - and have been interpreted as - short-wave instabilities in the tip vortices. As detailed in the paper, this instability mechanism has been observed experimentally in helical tip vortices and is caused by strain forces on the vortex core that are generated by neighboring vortices. In the absence of other external disturbances, this mechanism appears to be the primary instability leading to tip vortex breakdown. Because this instability involves disturbances at wavelengths similar to the vortex core size, the latter must be resolved adequately. Given the size of the vortex core in the ALM simulations, and the mesh size

in the current LES simulations, we argue that this is indeed the case. All appropriate references are provided in the amended manuscript. We admit, however, that the observed mechanisms may not be observed in reality, at least not to the same extent, as the tip vortex core is larger in the ALM simulations than in the experiments. In addition, as the Reviewer and some recent literature have pointed out, the cartesian grid may introduce small numerical perturbations, which could exacerbate the observed phenomena. We have pointed out these limitations clearly in the manuscript. We also believe that the fact that the primary instability mechanism changes when turbulence is included in the flow solver is indeed significant to this study and further highlights the importance of including such free-stream disturbances when studying FOWT wakes. We are confident that the Reviewer's concern is now more appropriately addressed.

**Specific comments:**

1. (Old comment #6) The name of the solver is now included at the beginning of section 3 and some references were provided. However, there is no description of the code and the references do not provide technical information about it. The only reference to (Richards et al., 2024) that I could find was a website. It was not properly referenced as a website in the manuscript, but I could not find an academic paper or technical report with the information provided in the references. If (Richards et al., 2024) refers to a technical publication, please provide more details so the reader may find it. From my point of view, a simple reference to a general website without technical details of the method is not sufficient for an article in an academic journal such as Wind Energy Science, which is a timeless registration of the work. Without the details of the numerical method, the work is not reproducible. Please, include a brief description of the numerical methods used by the solver and provide references that describe them in detail.

We generally agree with the Reviewer's concern that enough detail should be given to be able to independently verify this work. In our first draft we focused on providing enough details to fully describe the ALM model. The settings of the flow-solver are often solver-dependent and set-up dependent, as details as the turbulence closure, computational grid and equation solution approach all influence them. Based on the Reviewer's request, we included additional details regarding the numerical schemes that were used in the URANS and LES approaches in sections 3, 3.1, 3.2, and 3.3. Since CONVERGE is a closed-source CFD code, the most appropriate reference to the flow-solution approach is the solver manual. Therefore, this reference was added, and a disclaimer was made about the possibility of asking it to the industrial authors. In addition, we also included some additional references where the solver is applied to wind turbine solution problems. In summary, in the revised version, details regarding the flow solver, CFD settings and boundary condition specification can be found in sections 3 and 3.1, details regarding the computational grid are provided in sections 3.2 and Appendix B, details regarding the free-stream velocity fluctuation generation are provided in Appendix A, and all details regarding the ALM employed in this study can be found in section 3.3. Finally, details regarding the simulation length and statistical convergence of the results are provided in section 3.5. We would like to keep the narrative of this study as straight as possible to the point. In our opinion the manuscript is already heavy in details regarding the case set-up and solution method, and to avoid overburdening the reader we have not included the flow solution equations explicitly in the paper.

2. (Old comment #9) I congratulate the authors for the change from URANS-Hybrid to URANS-STG. I agree that the name is now adequate. However, the reference to the old name can be found in the introduction of section 4 ("h" stands for "hybrid"). Also, section 4.4 employs mostly the old nomenclature, which makes this section confusing and not consistent with the rest of the manuscript.

The authors would like to apologize for the confusion. The section has been revised and updated to reflect the new nomenclature. No reference to the old terminology should be present anymore in the final draft.

3. (Old comment #31) In the caption of Figure 31, it is not clear if \Delta WD is the amplitude of WD or the peak-to-peak amplitude. The text says "amplitude", which usually means peak-to-mean amplitude, but the formula is for peak-to-peak amplitude.

The Reviewer's comment is correct. The amplitude shown in the plots is calculated as $\Delta WD = (WD_{max} - WD_{min})/2$, where $WD_{max}$ and $WD_{min}$ are derived from the phase-averaged cycle, as already specified in the caption. A clarification has been added to the paper.

---

## Author Response (AR3)

**Response to the Editor**

The paper is still subject to 'major revisions' although it most likely is a minor issue that needs to be rectified (essentially a question of wordings regarding if the observed instabilities are numerical or physical). However, it is assessed by the reviewer to be of importance to phrase it correctly in order not to mislead unexperienced researchers.

Dear Editor, we have tried our best to revise the paper as suggested by the Reviewer. We would like to thank you again for the coordination of the review process of this study.

ooo   ooo   ooo

**Reviewer #2**

In the latest version of the manuscript "How does turbulence affect wake development in floating wind turbines? Some insights from comparative LES simulations and wind tunnel experiments", the authors satisfactorily addressed almost all of the comments. The authors show a clear effort to solve all of the comments, to improve the manuscript. The only open comment is still the one regarding whether the flow structures seen in the laminar simulations are created by numerical effects.

First of all, I would like to acknowledge that this is a good paper overall and it should be published in some form, if adequate steps are taken. Nevertheless, the open issue is a critical one and should be resolved. The best-case scenario would be the inclusion of results with improved discretization (or other additional simulations that could clarify this issue). From the authors' response, I understand that is not possible. In that case, I agree with the editor and the authors that the additional discussions could be an alternative to additional simulations. However, I am not convinced by the new discussion introduced on page 17:

- I do not believe the grid resolution is sufficient to capture shortwave instabilities. The comparison of the tip vortex core with experiments and other simulations of (Cioni et al. 2013) makes no sense for the current simulation. As stated in (Cioni et al. 2013), the initial core size in ALM simulations is a function of the kernel size. Therefore, the vortex core size of the experiment or other simulations is not related to the vortex core size of the current simulations. With a ratio of Kernel function width and local mesh size equals to 2.4, the initial core is not believed to be well-resolved. A value of 2 for this ratio (comparable to 2.4) has been observed to not represent the core accurately by numerous studies (Shives & Crawford, Wind Energy, 2013; Meyer Forsting & Troldborg, TORQUE 2020; Kleine et al., JFM 2023). There is vortex core growth due to diffusion, however, it is unclear if the growth would compensate for the lack of initial resolution. For example, Ribeiro et al. (2025) used a ratio of 7, to represent the vortices well. Anyhow, this is dependent on the numerical method. If the authors have data that show that the vortex core is resolved enough to capture short-wave instabilities, I suggest that they show it.
- According to Ribeiro et al. (2025), "they have been shown to occur in reality, but are only numerically captured via blade-resolved, scale-resolved simulations". I cannot be sure that they cannot be numerically captured via ALM. However, I have never seen a reliable paper that confirmed they are. I have seen instances where structures created by the ALM were misidentified as shortwave instabilities, but were then revealed to be something else.
- Even if the resolution and the method could capture shortwave instabilities, I cannot agree with the similarities the authors claim between the structures in Figure 10 and shortwave instabilities. For example, in figure 10(a), these "instabilities" are practically aligned in the direction of the flow, for different helices. In other words, the position z/D of the structure in one vortex matches perfectly the position z/D of a structure in the subsequent vortex. For me, it seems extremely unlikely that a phenomenon that is dominated by the dimensions of the vortex core would align so perfectly with the rotation period. From the preliminary simulations I have seen and run in the past, these structures look very similar to structures created by numerical effects related to the mesh. Unfortunately, I cannot provide a reference, because these were all preliminary simulations. The only reference I could find was: "Kleusberg E, Schlatter P, Henningson DS. Parametric study of the actuator line method in high-order codes. Technical report, KTH, 2017", which is not available online. It is available as an annex to "Kleusberg, E., 2019. Wind-turbine wakes-Effects of yaw, shear and turbine interaction. PhD Thesis, KTH, 2019", if you can find a physical copy.
- I agree with the phrase "Therefore, the results presented herein for the laminar case may not exactly match an equivalent experiment."

Despite all that, the work is good, the discussions are relevant and the presentation is of high quality. I do not intend to impede the publication of an interesting paper. However, in its current form, unless I am mistaken, I believe it could misdirect an inexperienced researcher such as an MSc or recent PhD student. Therefore, I suggest two possible paths:

1) The original suggestion, that further simulations be performed, to verify if these structures are created by numerical effects; or

2) That a discussion be added to the paper, in which it becomes very clear to the reader that the main hypothesis for the appearance of these structures is due to numerical effects.

If the second option is followed, I believe one way to carry out that would be:

- To acknowledge that the mechanisms for the creation of the observed flow structures were not identified;
- To indicate, very clearly, that the leading hypothesis is numerical errors;
- You may choose to disclose that the main hypothesis of numerical errors could not be confirmed by better-resolved simulations due to the costs of numerical simulations;
- To caution the reader that some of the numerical results might differ from experiments because of it;
- To reinforce that the analysis might still be relevant, even though it might differ from real flows, because it could give insights regarding best practices for numerical simulations;
- If shortwave instabilities are mentioned, my recommendation is that it becomes clear to the reader that this is an unlikely hypothesis (unless the authors can supply relevant data to support it) or that they are probably numerical shortwave instabilities that are not necessarily related to physical shortwave instabilities of vortices;
- To review the other sections to verify if this limitation would require changes.

We appreciate the additional details included by the Reviewer and the well-motivated scientific criticism that allowed a further critical analysis of our work. The paper has been revised according to the recommendations provided by the Reviewer at point #2 of his/her notes. We also checked that the message is conveyed coherently along the study.